# LipsNet++: Unifying Filter and Controller into a Policy Network

Xujie Song [1]   Liangfa Chen [2]   Tong Liu [1]   Wenxuan Wang [1]   Yinuo Wang [1]
Shentao Qin [1]   Yinsong Ma [3]   Jingliang Duan [1 2 *]   Shengbo Eben Li [1 *]

## Abstract

Deep reinforcement learning (RL) is effective for decision-making and control tasks like autonomous driving and embodied AI. However, RL policies often suffer from the action fluctuation problem in real-world applications, resulting in severe actuator wear, safety risk, and performance degradation. This paper identifies the two fundamental causes of action fluctuation: observation noise and policy non-smoothness. We propose LipsNet++, a novel policy network with Fourier filter layer and Lipschitz controller layer to separately address both causes. The filter layer incorporates a trainable filter matrix that automatically extracts important frequencies while suppressing noise frequencies in the observations. The controller layer introduces a Jacobian regularization technique to achieve a low Lipschitz constant, ensuring smooth fitting of a policy function. These two layers function analogously to the filter and controller in classical control theory, suggesting that filtering and control capabilities can be seamlessly integrated into a single policy network. Both simulated and real-world experiments demonstrate that LipsNet++ achieves the state-of-the-art noise robustness and action smoothness. The code and videos are publicly available at https://xjsong99.github.io/LipsNet_v2.

## 1. Introduction

Deep reinforcement learning (RL) has become a powerful approach for addressing optimal control tasks in physical environments (Bertsekas, 2019; Guan et al., 2021; Li, 2023; Wang et al., 2024b). Neural networks, capable of modeling complex nonlinear functions (Hornik et al., 1989; Kidger & Lyons, 2020), are commonly used as the container for the control policy fitted by RL. However, RL-trained policy networks often encounter the action fluctuation problem, where consecutive actions exhibit significant variations despite minor differences in the adjacent observations. While this problem is often overlooked during simulation and training stages, it will result in serious issues in real-world applications like performance reduction, actuators' wear, and safety risk (Song et al., 2023; Mysore et al., 2021; Chen et al., 2021; Wang et al., 2024a; 2025). This problem is prevalent in various scenarios, including drone control (Mysore et al., 2021; Shi et al., 2019), robot locomotion (Zhang et al., 2024), robot manipulation (Yu et al., 2021), and autonomous driving (Cai et al., 2020; Chen et al., 2021; Wasala et al., 2020; Lee et al., 2024), etc. These reported action fluctuation problem indicates that it constitutes a considerable gap in sim-to-real RL applications.

In order to make RL more applicable in real-world scenarios, researchers are working hard to solve the problem. CAPS (Mysore et al., 2021) and L2C2 (Kobayashi, 2022) mitigate fluctuations by introducing penalty terms in the actor loss, enforcing action similarity across successive time steps or similar states. $SR^2L$ (Shen et al., 2020; Zhao et al., 2022) employs adversarial noise and minimizes action differences between actual and perturbed states. PIC (Chen et al., 2021) and TAAC (Yu et al., 2021) design two-stage policies by using one network to output the current action, and the other to output action inertia scalar or make choice between the current and the last action. MLP-SN (Takase et al., 2020) and LipsNet (Song et al., 2023) smooth control actions by constraining the Lipschitz constant of policy network. However, both CAPS and L2C2 require sensitive hyperparameter tuning and involve compulsory sampling of neighboring states. $SR^2L$, PIC, and TAAC need special policy evaluation or policy improvement mechanisms. MLP-SN suffers from severe performance loss and difficulty in Lipschitz constant tuning. LipsNet, the previous state-of-the-art (SOTA) method, has network structure limitation and is unsuitable for high-real-time tasks due to slow inference speed. Most importantly, none of them explicitly address observation noise—a factor

---

[1]School of Vehicle and Mobility & College of AI, Tsinghua University, Beijing, China [2]School of Mechanical Engineering, University of Science and Technology Beijing, Beijing, China [3]Laboratory for Computational Sensing and Robotics, Johns Hopkins University, Baltimore, USA. Correspondence to: Shengbo Eben Li <lishbo@tsinghua.edu.cn>, Jingliang Duan <duanjl@ustb.edu.cn>.

*Proceedings of the 42nd International Conference on Machine Learning*, Vancouver, Canada. PMLR 267, 2025. Copyright 2025 by the author(s).

that our study identifies as a key cause of action fluctuation. In summary, achieving smooth control actions in a manner that is effective, simple, and broadly applicable across RL algorithms remains an open challenge.

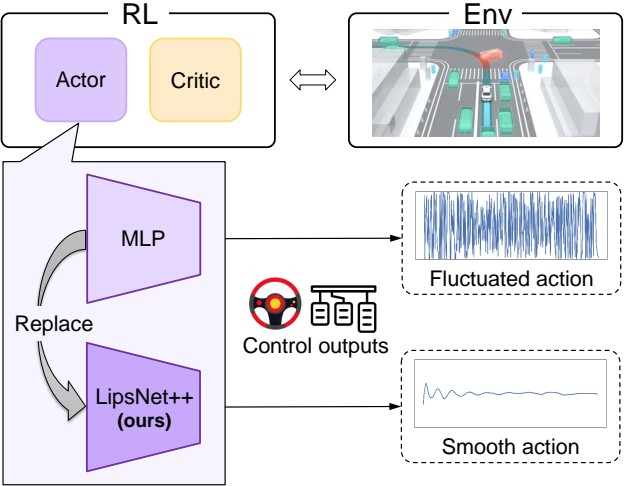

Figure 1: **LipsNet++ outputs smooth action.**

In this paper, we proposed a novel policy network structure, **LipsNet++**, achieving action smoothing in RL effectively, simply and flexibly. We identified the fundamental reasons for causing action fluctuation are the non-smoothness of policy network and the existence of observation noise. LipsNet++ adopts two corresponding network modules to explicitly tackle them at the same time. Firstly, we propose a Fourier filter layer to filter observation noise. In this layer, fast Fourier transform (FFT) is used to obtain the frequency features of sequential observations, and a trainable filter matrix automatically extracts important frequencies and suppress noise frequencies in observations. Secondly, a Lipschitz controller layer is introduced, whose Lipschitz constant is constrained by a proposed Jacobian regularization technique. The controller layer can be arbitrary derivable network structures, breaking the network structure limitation of LipsNet who only supports multilayer perceptron (MLP) with piecewise linear activation functions. This layer no longer requires the computation of the Jacobian matrix during inference, thereby breaking the inference speed limitation of LipsNet. The Fourier filter layer and Lipschitz controller layer function analogously to the filter and controller in classical control theory, respectively, suggesting that filtering and control capabilities can be seamlessly integrated into a single policy network.

**Experiment results.** Simulated and physical experiments verify that LipsNet++ has achieved the SOTA performance. For the simulated tasks, we conduct experiments on the double integrator environment and DeepMind control suite benchmark (DMControl). For example, in DMControl's walker environment, LipsNet++ increases the total average

return (TAR) by 3.4% and reduces the action fluctuation ratio (AFR) by 35.5% compared to LipsNet (Song et al., 2023), which is the previous SOTA network. Additionally, an experiment on physical mini-vehicles is implemented for real-world testing, where the vehicle is going to track given trajectories and avoid moving obstacle under various noise levels. Results show that LipsNet++ increases the TAR by 5.8% and reduces the AFR by 90.0% compared to MLP.

**Technical contributions.** LipsNet++ is a novel network, addressing the action fluctuation problem in the real-world applications of RL. Our contributions are four-fold: **(1)** We identify the two fundamental reasons that cause action fluctuation, and propose a policy network LipsNet++ to explicitly tackle the two reasons in a decoupled manner; **(2)** We propose a Fourier filter layer in LipsNet++, capable of automatically extracting valuable frequencies while suppressing noise frequencies in observations; **(3)** We propose a Lipschitz controller layer with Jacobian regularization technique to enhance the smoothness of policy fitting, which breaks LipsNet's structure limitation and inference speed limitation; **(4)** LipsNet++ suggests that filtering and control capabilities can be seamlessly integrated into a single policy network, providing valuable insights for future policy network design. The code is released to facilitate the implementation and future research.

## 2. Preliminaries

### 2.1. Actor-Critic Reinforcement Learning

Actor-critic method, consisting of an actor network and a critic network as shown in Figure 1, forms the backbone of many RL algorithms. The actor network fits a policy $\pi : \mathcal{S} \to \mathcal{A}$ that mapping from state space to action space. Therefore, the actor network is also called as policy network. The goal of RL is to train a policy $\pi$ maximizing the expected return:

$$J_\pi = \mathbb{E}_{\tau \sim \rho_\pi} \left[ \sum_{t=0}^{T} \gamma^t r_t \right],$$

where $\rho_\pi$ is the distribution of state-action trajectory induced by policy $\pi$, $T$ is the termination time of an episode, $0 \leq \gamma \leq 1$ is the discount factor, and $r_t$ represents the reward. The critic network fits a value function $V(s)$ or $Q(s, a)$, mapping from the state-action pairs to the expected returns, to evaluate the actions taken by the actor.

In policy evaluation phase, the critic is updated by minimizing the temporal difference (TD) error. For example, the Q-value network in DDPG (Lillicrap et al., 2015) parameterized by $\varphi$ is updated by

$$\min_{\varphi} \mathbb{E}_{s,a,r,s' \sim \mathcal{D}} \left[ \left( Q_\varphi(s, a) - r - \gamma Q_{\varphi_{\text{targ}}}(s', a') \right)^2 \right],$$

where $\mathcal{D}$ is the replay buffer, $s'$ is the next state, $a'$ is the next action obtained by the target actor network, and $Q_{\varphi_{\text{targ}}}$ is the return estimated by the target critic network.

In policy improvement phase, the actor is updated by maximizing the expected return predicted by the critic. Taking DDPG as an example again, the actor network is updated by minimizing the actor loss function:

$$\mathcal{L} = \mathbb{E}_{s \sim \mathcal{D}} \left[ -Q_\varphi(s, \pi(s)) \right].$$

### 2.2. Action Fluctuation Ratio

Action fluctuation ratio (AFR) is an index to quantitatively measure the fluctuation level of control actions (Chen et al., 2021; Song et al., 2023). It is defined as

$$\xi(\pi) = \mathbb{E}_{\tau \sim \rho_\pi} \left[ \frac{1}{T} \sum_{t=1}^{T} ||a_t - a_{t-1}|| \right],$$

where $\rho_\pi$ is the distribution of state-action trajectory induced by policy $\pi$, $T$ is the termination time of episodes, $a_t$ and $a_{t-1}$ are two adjacent actions, and $||\cdot||$ is the norm of action difference vector [1].

Beside the total average return (TAR), AFR is also an important indicator to evaluate policies' performance in the real world. The smaller AFR is, the smoother action sequence policy $\pi$ has.

## 3. Method

### 3.1. Reasons Identification of Action Fluctuation

To ensure that RL agents produce smooth actions, it is necessary to first identify the root cause of action fluctuation. In decison-making and control tasks, the actions are calculated by the policy network $\pi$ according to the current observation $o_t$, i.e. $a_t = \pi(o_t)$. And the current observation $o_t$ is composed by the current state $s_t$ and observation noise $\sigma_t$, i.e. $o_t = s_t + \sigma_t$. The rate of action change over time is $\frac{\mathrm{d}a_t}{\mathrm{d}t} = \frac{\mathrm{d}\pi(o_t)}{\mathrm{d}o_t} \cdot \frac{\mathrm{d}o_t}{\mathrm{d}t}$, then we can derive that

$$\left\| \frac{\mathrm{d}a_t}{\mathrm{d}t} \right\| \leq \left\| \frac{\mathrm{d}\pi(o_t)}{\mathrm{d}o_t} \right\| \cdot \left( \left\| \frac{\mathrm{d}s_t}{\mathrm{d}t} \right\| + \left\| \frac{\mathrm{d}\sigma_t}{\mathrm{d}t} \right\| \right). \quad (1)$$

To mitigate action fluctuation, $\left\| \frac{\mathrm{d}a_t}{\mathrm{d}t} \right\|$ must be controlled within a reasonable range. From Equation (1), we know $\left\| \frac{\mathrm{d}a_t}{\mathrm{d}t} \right\|$ is affected by three parts: a red term of policy derivative, reflecting the level of policy smoothness; a blue term of noise change rate, reflecting the level of observation noise; and an inherent derivative term $\left\| \frac{\mathrm{d}s_t}{\mathrm{d}t} \right\|$ of the target dynamics system which cannot be modified.

---

[1] Throughout the paper, $||\cdot||$ denotes the 2-norm of a vector or a matrix.

Based on the above analysis, the two fundamental causes of action fluctuation can be identified: (1) the non-smoothness of policy network, and (2) the existence of observation noise, corresponding to the red and blue terms, respectively.

**Non-smoothness of policy network.** A non-smooth policy network means that RL fits a non-smooth policy function mapping from the states to control actions. The mapping function has significant output differences even if the inputs are closely adjacent. Consequently, when the state changes with time, a non-smooth action sequence is generated. Appendix B visualizes the effect of a non-smooth policy.

**Existence of observation noise.** The noise results in the discontinuous changes in observations, making the actions produced by the policy network at the adjacent time stamps erratically differ. Even if the policy function fitted by the policy network is smooth enough, actions can still be fluctuated because of the erratic observation noise.

Therefore, achieving sufficiently smooth control actions requires addressing both fundamental reasons simultaneously. However, previous works have neither clearly identified these two causes nor considered them together. While some studies acknowledge the impact of observation noise, they attempt to enhance robustness by reducing the Lipschitz constant of the policy network (Takase et al., 2020; Song et al., 2023), i.e. improving the smoothness of policy network, rather than directly filtering the observation noise. Such a non-decoupled approach results in actions being insufficiently smooth, and performance degradation when higher action smoothness is required. In this paper, we introduce the Fourier filter layer and Lipschitz controller layer to explicitly and decoupledly address these two fundamental causes, providing a more effective and principled solution.

### 3.2. Fourier Filter Layer

Fourier Transform is a widely used frequency analysis tool, which can also be employed in neural networks for feature extraction (Lee-Thorp et al., 2022; Rao et al., 2021; Tan et al., 2024). To mitigate the action fluctuation caused by observation noise, we propose the Fourier filtering layer based on fast Fourier Transform. The workflow of Fourier filter layer is shown in Figure 2.

Given $N$ historical observations $o_t, o_{t-1}, \cdots, o_{t-N+1} \in \mathbb{R}^D$ where $D$ denotes the dimension of features, the Fourier filter layer concatenates them as a matrix $x \in \mathbb{R}^{N \times D}$, and calculates the frequency feature matrix $X \in \mathbb{C}^{N \times D}$ using 2D discrete Fourier transformation:

$$X_{u,v} = \sum_{n=0}^{N-1} \sum_{d=0}^{D-1} x_{n,d} \cdot e^{-j2\pi\left(\frac{un}{N} + \frac{vd}{D}\right)}, \quad (2)$$

where $x_{n,d}$ denotes the $d$-th feature of the $n$-th observation signal, $X_{u,v}$ denotes the element located at the $u$-th row

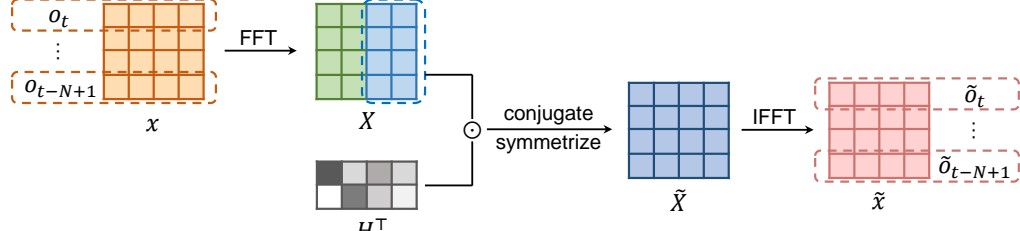

Figure 2: **Workflow of Fourier filter layer.** Firstly, FFT converts historical observations to frequency feature matrix $X$. Then, half of $X$ is multiplied by a trainable filter matrix $H$, and a complete matrix $\tilde{X}$ is generated by conjugate symmetrizing. Finally, IFFT converts $\tilde{X}$ to filtered time-domain signals.

and $v$-th column of the frequency feature matrix $X$, and $j$ represents the imaginary unit. When the length of historical observations is less than $N$, the missing parts are padded with 0. In FFT, Zero-padding does not alter the primary frequency components of the signal, and it merely increases the spectral resolution (Jung et al., 2019). The magnitude of $X_{u,v}$ denotes the signal intensity at the frequency combination $(u, v)$, where $u$ and $v$ are frequency indices rather than the actual frequency values. Since the observations only consist of real values, the resulting matrix $X$ exhibits conjugate symmetry, i.e. $\overline{X_{u,v}} = X_{N-u,D-v}$. It means that half of $X$ could represent the complete information contained in the signal.

Then, half of $X$, denoted as $X_{\text{half}} \in \mathbb{C}^{N \times \lfloor \frac{D}{2} \rfloor + 1}$, is subjected to a Hadamard product with a trainable filter matrix $H \in \mathbb{C}^{N \times \lfloor \frac{D}{2} \rfloor + 1}$. After that, a complete matrix $\tilde{X} \in \mathbb{C}^{N \times D}$ is restored by conjugate symmetrizing the product matrix:

$$\tilde{X} = \text{symmetrize}(X_{\text{half}} \odot H). \tag{3}$$

By choosing $H$ as a complex matrix instead of real matrix, the Fourier filtering layer can not only alter frequency amplitudes but also perform feature extraction. The magnitudes of the elements in $H$ determine which frequency is suppressed or strengthened. To enable the noise filtering capability of policy network, we encourage the magnitudes of elements in $H$ to be as small as possible. In this way, policy network can automatically extract valuable frequencies and filter out less relevant frequencies where noise may exist. The tailored actor loss becomes

$$\mathcal{L}' = \mathcal{L} + \lambda_h \|H\|_F, \tag{4}$$

where $\|H\|_F$ is the Frobenius norm of $H$, and $\lambda_h$ is a hyperparameter coefficient.

Finally, the resulted frequency feature matrix $\tilde{X}$ is recovered to the time-domain signals by 2D inverse discrete Fourier transformation:

$$\tilde{x}_{n,d} = \frac{1}{ND} \sum_{u=0}^{N-1} \sum_{v=0}^{D-1} \tilde{X}_{u,v} \cdot e^{j2\pi\left(\frac{un}{N} + \frac{vd}{D}\right)}. \tag{5}$$

Because $\tilde{X}$ is a conjugate symmetric matrix, the matrix $\tilde{x} \in \mathbb{R}^{N \times D}$ becomes a real matrix. By slicing rows from the matrix $\tilde{x}$, the filtered features $\tilde{o}_t, \tilde{o}_{t-1}, \cdots, \tilde{o}_{t-N+1} \in \mathbb{R}^D$ are obtained. The signal $\tilde{o}_t$, representing the filtered feature corresponding to the current timestamp, is selected as the input for the subsequent Lipschitz controller layer.

### 3.3. Lipschitz Controller Layer

**Definition 3.1** (Local Lipschitz Constant). Suppose $f : \mathbb{R}^n \to \mathbb{R}^m$ is a continuous neural network. The $K(x)$ is defined as the local Lipschitz constant of $f$ on the neighborhood of $x$:

$$K(x) = \max_{x_1, x_2 \in \mathcal{B}(x,\rho)} \frac{\|f(x_1) - f(x_2)\|}{\|x_1 - x_2\|}, \tag{6}$$

where $\mathcal{B}(x,\rho)$ denotes the open ball area with radius $\rho > 0$ centered at the point $x$ in the Euclidean space, i.e. $\mathcal{B}(x,\rho) = \{x' : \|x' - x\| < \rho\}$.

Lipschitz constant characterizes the landscape smoothness of a function. By viewing the policy network as a mapping function from states to actions, Lipschitz constant can reflect the smoothness of the policy function. A lower Lipschitz constant means a smoother policy function, leading to smoother actions (Ames et al., 2016; Kobayashi, 2022; Song et al., 2023; Takase et al., 2020). MLP-SN (Takase et al., 2020) constrains the Lipschitz constant by applying spectral normalization (SN) (Miyato et al., 2018) on each layer of policy network. However, it leads to tuning difficulty and severe performance loss, because the desired network-wise Lipschitz continuity is realized by layer-wise Lipschitz constraints (Bhaskara et al., 2022; Wu et al., 2021). Instead, LipsNet (Song et al., 2023) proposes a network-wise method, Multi-dimensional Gradient Normalization (MGN). However, LipsNet is not applicable in high-real-time tasks due to the Jacobian matrix calculation during forward inference, and its network structure is limited to MLP with piecewise linear activation functions.

To overcome the above challenges, we propose the Jacobian regularization method to conveniently reduce the Lipschitz

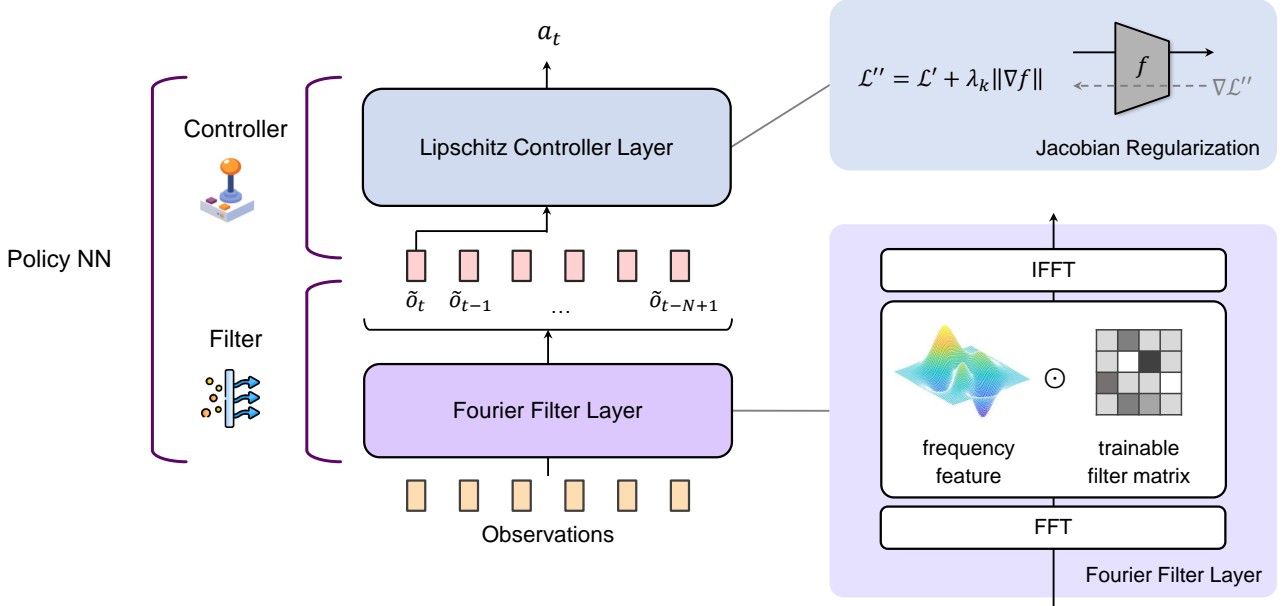

Figure 3: **Overall structure of LipsNet++.** Historical observations are processed by Fourier filter layer, where a trainable filter matrix is used for frequency selection. The filtered feature $\tilde{o}_t$ is inputted into Lipschitz controller layer whose Lipschitz constant is constrained by a Jacobian regularization. The parameters in LipsNet++ are updated by tailored actor loss $\mathcal{L}''$.

constant of our controller layer. The Jacobian norm is a commonly used index of function smoothness and robustness (Hoffman et al., 2019; Lee et al., 2023).

**Theorem 3.2** (Lipschitz's Jacobian Approximation). *Let $f : \mathbb{R}^n \to \mathbb{R}^m$ be a continuously differentiable neural network. The Jacobian norm $\|\nabla_x f\|$ is an approximation of $f$'s local Lipschitz constant within neighborhood $\mathcal{B}(x, \rho)$, centered at $x$ with radius $\rho$. The approximation error is*

$$\max_{\delta \in \mathcal{B}(0,\rho)} \left[ \left( \nabla_x \|\nabla_x f(x)\| \right)^\top \delta + o(\delta) \right],$$

*where $o(\delta)$ is a higher-order infinitesimal term with respect to $\delta$. Moreover, as $\rho \to 0$, the Jacobian norm converges to the exact local Lipschitz constant, i.e.*

$$\lim_{\rho \to 0} \|\nabla_x f\| = K(x).$$

*Proof.* See Appendix A in the supplementary material. □

We know that Lipschitz constant reflects policy's landscape smoothness (Takase et al., 2020; Song et al., 2023) and Theorem 3.2 proves that Jacobian norm is an approximation of the local Lipschitz constant, therefore we can conveniently enhance the policy smoothness by reducing Jacobian norm. The actor loss is tailored from $\mathcal{L}'$ in Equation (4) into

$$\mathcal{L}'' = \mathcal{L}' + \lambda_k \|\nabla f\|, \tag{7}$$

where $f$ means the Lipschitz controller layer and $\lambda_k$ is a constant coefficient. The proposed Jacobian regularization

is superior to the Lipschitz constraint methods used in MLP-SN and LipsNet because: (1) It is a network-wise rather than layer-wise constraint method, avoiding severe performance loss; (2) It does not require a predefined initial/target Lipschitz constant, avoiding the tuning difficulty; (3) It does not need to calculate Jacobian matrix during forward inference, making it applicable in high-real-time tasks; (4) It is suitable for arbitrary derivable network structures.

The overall structure of LipsNet++ is shown in Figure 3. The pseudocode of LipsNet++ is illustrated in Appendix C.

### 3.4. User-friendly Packaging

We have packaged LipsNet++ as an user-friendly PyTorch (Paszke et al., 2019) module and released the code. When network's backward propagation is called, a backward hook function will awake to automatically replace the gradient $\nabla\mathcal{L}$ by $\nabla\mathcal{L}''$. In this way, LipsNet++ can be easily integrated into almost all actor-critic RL algorithms like DDPG (Lillicrap et al., 2015), TD3 (Fujimoto et al., 2018), PPO (Schulman et al., 2017), TRPO (Schulman et al., 2015), SAC (Haarnoja et al., 2018) and DSAC (Duan et al., 2025). As shown below, practitioners can use it as simple as MLP.

```
net = LipsNet++()
out = net(input)
...
loss.backward()
```

# 4. Experiments

In this section, we comprehensively evaluate LipsNet++ with model-based, model-free, deterministic, and stochastic RL algorithms in both simulated and real-world tasks.

## 4.1. Double Integrator

Double integrator is a classic linear quadratic control task, which is commonly used to test the performance of controllers. In the environment, a particle is moving along an axis without resistance (Song et al., 2023). The observations include position $x$ and velocity $v$ of the particle. The control action is particle acceleration $a$ that parallel to the axis. A schematic diagram of the environment is shown in Figure 4.

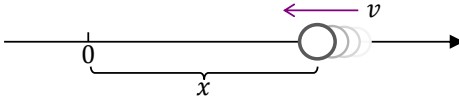

Figure 4: **Double integrator task.** A particle is moving along an axis without resistance.

The reward function is $r = -2x^2 - v^2 - a^2$, which incentives the particle to remain stable at the origin, i.e. $x = 0, v = 0, a = 0$. The Infinite-time Approximate Dynamic Programming (INFADP) (Li, 2023), a model-based RL algorithm, is used for train without noise. When testing policy networks, the particle has nonzero initial position and velocity, and the noise for each observation dimension is distributed in $U(-0.2, 0.2)$. More details and hyperparameters are shown in Appendix D.

The results are presented in Figure 6 and 7. In Figure 6(a), 30 episodes are simulated starting from the same initial state. The solid line and shadow area respectively denote the mean and standard deviation of actions. The shadow areas imply the action fluctuation amplitude of LipsNet++ is much smaller than that of MLP, and is on par with LipsNet. Figure 6(b) depicts action trajectories for a single episode, which reveals that LipsNet++ has better action continuity than LipsNet under the same level of action fluctuation amplitude. This conclusion is confirmed again by Figure 6(c), where action trajectories are decomposed by FFT and the action frequency induced by LipsNet++ is shown to be more distributed in the low-frequency range.

To further evaluate LipsNet++, we set different observation noise amplitudes and compare with previous works. As Figure 7(a) shows, when noise increases, LipsNet++ maintains the highest TAR and its TAR declines at the slowest rate. As Figure 7(b) shows, when noise increases, LipsNet++ maintains the lowest AFR and its AFR grows at the slowest rate. We then compare the performance in high-noise environment, i.e. noise amplitude is 0.3. Compared to LipsNet-L, the previous SOTA network, LipsNet++ achieves an 8.2%

increase in TAR and a 75.0% reduction in AFR. Therefore, LipsNet++ achieves a new SOTA performance with a significant advantage in action smoothness.

Furthermore, an ablation study for the two techniques in LipsNet++ is implemented in Appendix E, the sensitivity analysis for hyperparameters $\lambda_k$ and $\lambda_k$ is provided in Appendix F, and the sensitivity analysis for hyperparameter $N$ is provided in Appendix G. Additionally, policy networks' computational efficiency are evaluated in Appendix H, including the time usages of forward and backward propagations. Based on all the results, a performance radar chart is depicted in Figure 7(c), which implies the overall performance of LipsNet++ is much better than previous works.

## 4.2. DeepMind Control Suite

The DeepMind Control Suite (DMControl) (Tassa et al., 2018) consist of several well-designed continuous control tasks. Currently, it stands as one of the most recognized benchmarks in the fields of RL and continuous control (Mu et al., 2022). In this paper, we focus on four of its environments: Cartpole, Reacher, Cheetah, and Walker. The visualization of these environments are shown in Figure 5, and more information are described in Appendix I.

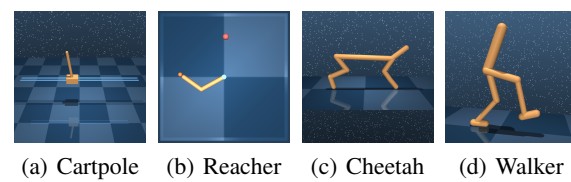

(a) Cartpole    (b) Reacher    (c) Cheetah    (d) Walker

Figure 5: **DeepMind control suite benchmark.** Four environments in DeepMind control suite are selected for testing.

We employ the Twin Delayed Deep Deterministic Policy Gradient (TD3) (Fujimoto et al., 2018), a model-free RL algorithm, for training. The hyperparameters for TD3 remain consistent across all environments, except for the coefficients $\lambda_k$, $\lambda_h$, and the length of historical observations $N$. All hyperparameters are listed in Appendix J. To evaluate comprehensively, networks are tested on both noise-free and noisy environments. Figure 9 visualizes the results in noisy environments. The learned filter matrix $H$ is visualized in Figure 24 to show the noise filtering ability of LipsNet++. All results are summarized in Table 11 and 12, from which we can find that LipsNet++ has the highest TAR and the lowest AFR in all cases. For example, LipsNet++ increases the TAR by 3.4% and reduces the AFR by 35.5% in Walker environment compared to LipsNet, which is the previous SOTA network. Appendix K shows a comparison in Cartpole environment between LipsNet++ and reward penalty method. All these results imply that LipsNet++ has perfect action smoothness and noise robustness.

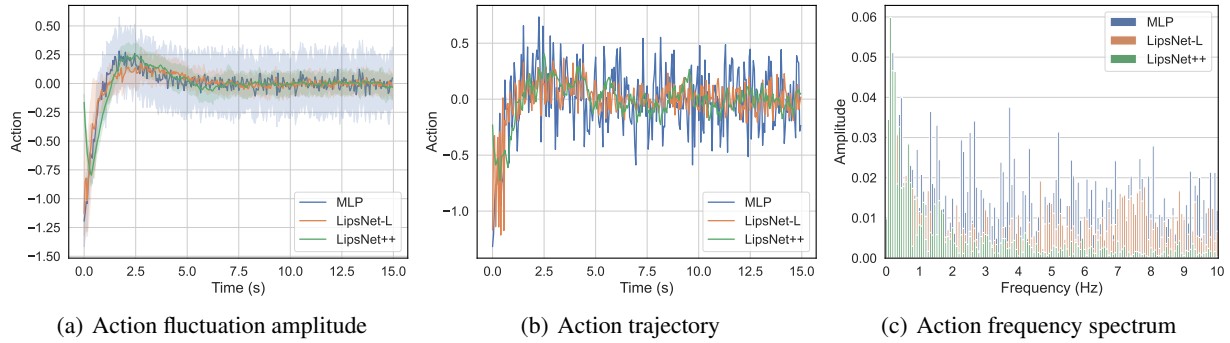

(a) Action fluctuation amplitude

(b) Action trajectory

(c) Action frequency spectrum

Figure 6: **Action visualization in double integrator environment.** (a) The action fluctuation amplitude of LipsNet++ is smaller than that of MLP, and is on par with LipsNet. (b) LipsNet++ has better action continuity than MLP and LipsNet. (c) LipsNet++'s action frequency is more distributed in the low-frequency range.

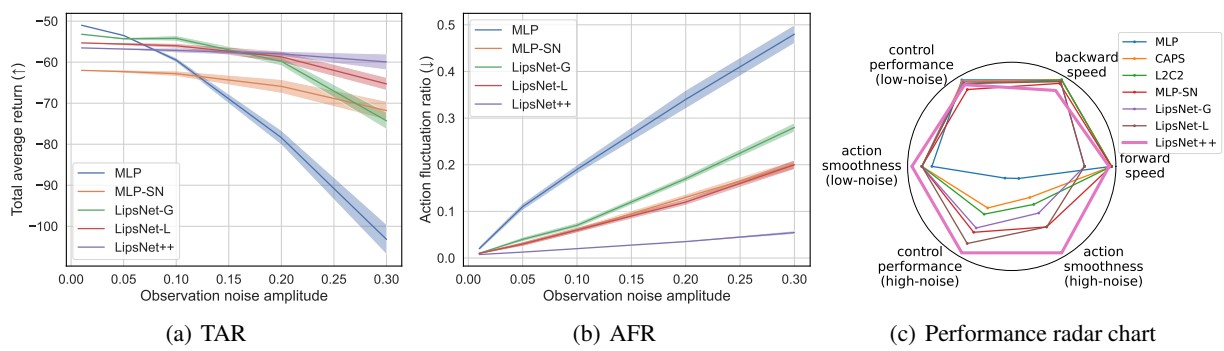

(a) TAR

(b) AFR

(c) Performance radar chart

Figure 7: **Performance comparison in double integrator environment.** (a) The TAR of LipsNet++ declines at the slowest rate when noise increases. (b) The AFR of LipsNet++ grows at the slowest rate when noise increases. (c) LipsNet++ has the best overall performance in the evaluation metrics of control performance, action smoothness, and forward speed, etc.

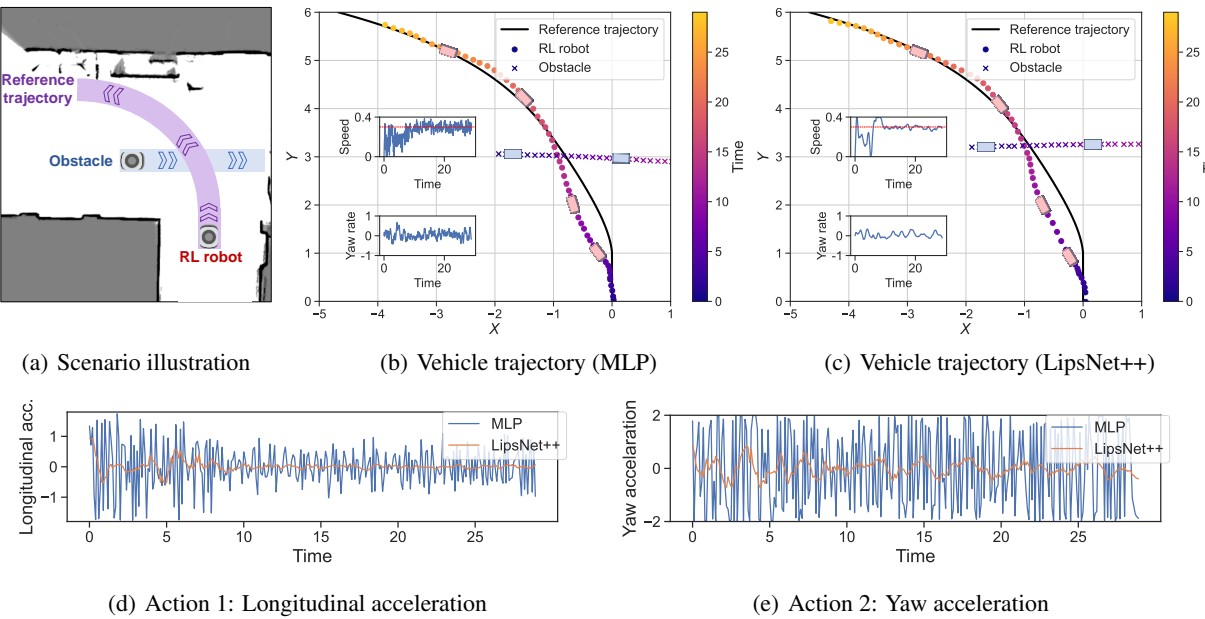

(a) Scenario illustration

(b) Vehicle trajectory (MLP)

(c) Vehicle trajectory (LipsNet++)

(d) Action 1: Longitudinal acceleration

(e) Action 2: Yaw acceleration

Figure 8: **Result of scenario 3.** The noise amplitude is 10. (a) The RL robot aims to turn left. (b,c) The vehicle states and trajectories produced by MLP and LipsNet++. (d,e) The control actions produced by MLP and LipsNet++. LipsNet++ produces much smoother control actions than MLP.

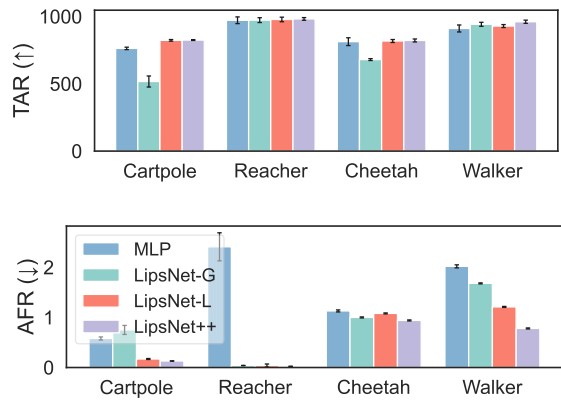

Figure 9: **Performance comparison in DMControl.** It shows the TAR and AFR in noisy environments. LipsNet++ has the highest TAR and the lowest AFR in all cases.

### 4.3. Mini-Vehicle Driving

Vehicle trajectory tracking is an important task in autonomous driving (Guan et al., 2022; Mu et al., 2020). To validate LipsNet++ in the real world, we conduct an experiment on physical vehicles. As Figure 26 shows, the vehicle moves by two differential wheels, aiming to track reference trajectory and velocity while avoiding obstacle. The observations and actions are listed in Table 15. We set up four diverse scenarios, as described in Table 1 and visualized in Figure 28. Detailed introduction of the vehicle, control mode, and scenarios are described in Appendix L. The Distributional Soft Actor-critic (DSAC) (Duan et al., 2025), a model-free RL algorithm, is used for training. The tests in all scenarios are accomplished by the same networks. For real-world highway vehicles, RL observations rely on perception results where sensor noise is amplified by perception algorithms. To precisely simulate such scenario, we assigned various noise amplitudes.

Table 1: **Scenario descriptions.**

| Scenario No. | RL robot | Obstacle robot |
|:---:|:---:|:---:|
| 1 | go straight | stationary |
| 2 | go straight | moving |
| 3 | turn left | moving |
| 4 | go straight | aggressive |

In scenario 3 with 10 times noise, the results are shown in Figure 8, its video snapshots are recorded in Figure 11. The RL robot successfully tracks the reference trajectory and avoids obstacle by slightly shifting to yield. As shown in Figure 8(d)(e), it is evident that LipsNet++ produces much smoother control actions than MLP. The smoother

actions result in smoother vehicle states, i.e. speed and yaw rate, which are shown in Figure 8(b)(c). These results consistently hold true across all scenarios, as illustrated in Appendx M. Furthermore, in scenario 4 with 10 times noise, the MLP-driven robot crashed while LipsNet++ successfully completed the task, as shown in Figure 48 and 49.

The learned filter matrix $H$ is visualized in Figure 12 to show the noise filtering ability of LipsNet++. Figure 12(a) and 30(b) show the frequency distributions of observation in noise-free and noisy environments, respectively. Figure 12(b) implies that the learned filter matrix mainly focus on the frequencies containing observation information, and rarely focus on the frequencies containing noises.

The average TAR and AFR for the first three scenarios are depicted in Figure 13. As Figure 13(a) shows, when noise increases, LipsNet++ maintains the highest TAR and its TAR declines much slower than MLP's. As Figure 13(b) shows, when noise increases, LipsNet++ maintains the lowest AFR and its AFR grows much slower than MLP's. In the high-noise environment (noise amplitude is 20), LipsNet++ achieves 5.9% increase in TAR and 90.0% reduction in AFR. More results are listed in Appendix M, implying LipsNet++ has much better action smoothness and noise robustness.

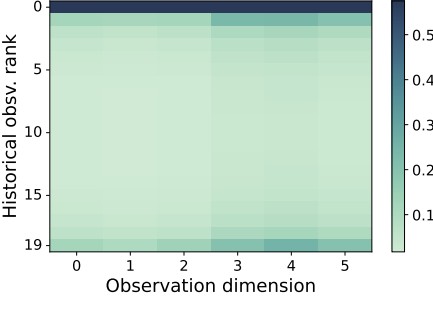

(a) Obsv. frequency (noise-free)

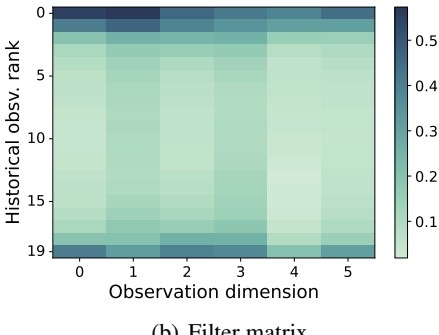

(b) Filter matrix

Figure 12: **Observation frequency and filter matrix.** The color in (a) and (b) means the frequency intensity of noise-free observations and the element magnitude in filter matrix. The matched color distribution implies LipsNet++ can extract important frequencies and filter out noise frequencies.

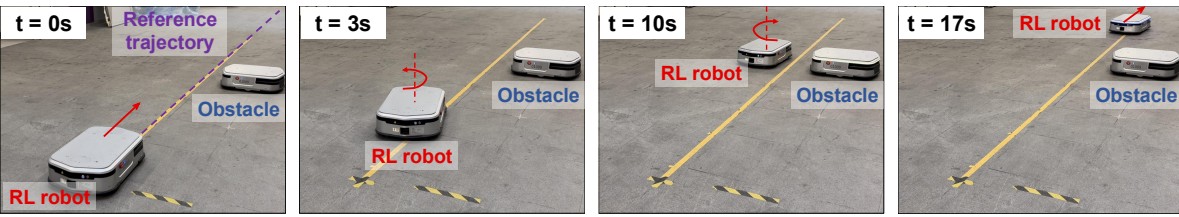

Figure 10: **Snapshots of scenario 1.** The RL robot first shifts left to navigate around the stationary obstacle robot, then shifts right to resume tracking the reference trajectory. All test videos are available at https://xjsong99.github.io/LipsNet_v2.

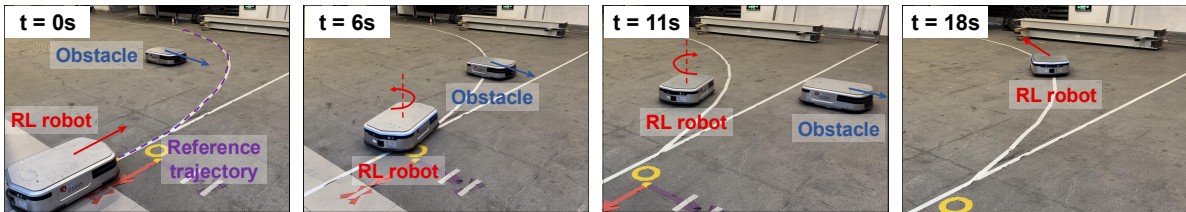

Figure 11: **Snapshots of scenario 3.** Video snapshots of Figure 8(c). The RL robot first shifts left to yield to the moving obstacle robot, then resumes tracking the left-turn reference trajectory. All videos: https://xjsong99.github.io/LipsNet_v2.

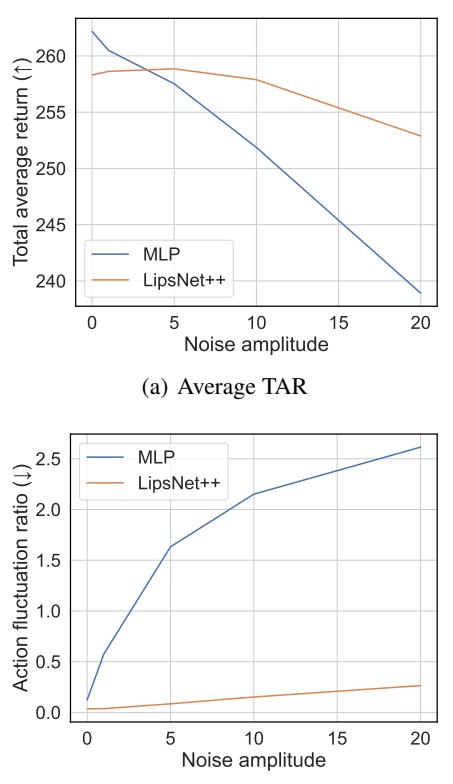

(a) Average TAR

(b) Average AFR

Figure 13: **Performance trend with increasing noise.** They show the average TAR and AFR of scenarios 1-3. (a) The TAR of LipsNet++ declines much slower than MLP's. (b) The AFR of LipsNet++ grows much slower than MLP's.

## 5. Conclusion

In this paper, we identify the two fundamental causes of action fluctuation and propose a novel policy network LipsNet++, where a Fourier filter layer and a Lipschitz controller layer are introduced to explicitly tackle the two causes. Fourier filter layer has observation noise filtering capability based on a learnable filter matrix. Lipschitz controller layer uses a Jacobian regularization for smoothly fitting policy functions, breaking LipsNet's structure limitation and inference speed limitation. LipsNet++ can be easily integrated into most actor-critic RL algorithms. Simulated and real-world experiments show that LipsNet++ has excellent action smoothness and noise robustness, achieving a new SOTA performance. The unifying of filtering and control capabilities providing valuable insights for future policy network design. We hope the research could make a contribution to the real-world applications of RL.

## Acknowledgements

This study is supported by National Key R&D Program of China with 2022YFB2502901, and Tsinghua University-Didi Joint Research Center for Future Mobility.

## Impact Statement

LipsNet++ has positive impacts on the AI community by effectively addressing the action fluctuation problem in RL. LipsNet++ breaks through the bottleneck of action fluctuation and poor robustness faced by RL, which accelerates

the process of RL's real-world applications. It mitigates the wear of actuators, safety risks, and performance reduction caused by fluctuated actions. The unifying of filtering and control capabilities providing valuable insights for future policy network design. LipsNet++ benefits many industrial fields, including drone control, decision-making and control of autonomous vehicles, and embodied AI.

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

## A. Theoretical Results

**Lemma A.1** (Equivalent Form of Lipschitz Constant). *Suppose $f : \mathbb{R}^n \to \mathbb{R}^m$ is a continuously differential neural network. Then its local Lipschitz constant $K(x)$ has an equivalent form besides equation (6):*

$$K(x) = \max_{x' \in \mathcal{B}(x,\rho)} \|\nabla f(x')\|. \tag{8}$$

*Proof.* We assume the local Lipschitz constant of $f$ over $\mathcal{B}(x, \rho)$ is $K_x$, which means $K_x = \max_{x_1, x_2 \in \mathcal{B}(x,\rho)} \frac{\|f(x_1) - f(x_2)\|}{\|x_1 - x_2\|}$.

**(a)** Firstly, we prove that $\|\nabla f(x')\| \leq K_x, \forall x' \in \mathcal{B}(x, \rho)$. Because the local Lipschitz constant is $K_x$, we know that

$$\|f(x_1) - f(x_2)\| \leq K_x \|x_1 - x_2\|, \ \forall x_1, x_2 \in \mathcal{B}(x, \rho). \tag{9}$$

Let $h(t) = f(x' + t \cdot v)$ where $x' \in \mathcal{B}(x, \rho)$, $t \in \mathbb{R}$, and $v \in \mathbb{R}^n$, then its first-order derivative function is $h'(t) = \nabla f(x' + t \cdot v) \cdot v$. From the *Newton-Leibniz formula*, we know

$$h(\alpha) - h(0) = \int_0^\alpha h'(t) \, dt,$$

which means

$$f(x' + \alpha \cdot v) - f(x') = \int_0^\alpha \nabla f(x' + t \cdot v) \cdot v \, dt.$$

By taking the 2-norm on both sides and considering the condition (9), we get

$$\left\| \int_0^\alpha \nabla f(x' + t \cdot v) \, dt \cdot v \right\| = \|f(x' + \alpha \cdot v) - f(x')\|$$
$$\leq \alpha K_x \|v\|.$$

Divide $\alpha$ on both sides then let $\alpha \to 0^+$, get

$$\|\nabla f(x') \cdot v\| \leq K_x \|v\|, \ \forall v.$$

From the definition of matrix norm, we know

$$\|\nabla f(x')\| = \max_{v \neq 0} \frac{\|\nabla f(x') \cdot v\|}{\|v\|} \leq K_x, \forall x' \in \mathcal{B}(x, \rho).$$

**(b)** Secondly, we prove that $\max_{x' \in \mathcal{B}(x,\rho)} \|\nabla f(x')\| \geq K_x$. Let $h(t) = f(x_1 + t(x_2 - x_1))$ where $t \in (0, 1)$ and $x_1, x_2 \in \mathcal{B}(x, \rho)$, then its first-order derivative function is $h'(t) = \nabla f(x_1 + t(x_2 - x_1)) \cdot (x_2 - x_1)$. From the *Newton-Leibniz formula*, we know

$$h(1) - h(0) = \int_0^1 h'(t) \, dt,$$

which means

$$f(x_2) - f(x_1) = \int_0^1 \nabla f(x_1 + t(x_2 - x_1)) \cdot (x_2 - x_1) \, dt$$
$$= \left( \int_0^1 \nabla f(x_1 + t(x_2 - x_1)) \, dt \right) (x_2 - x_1).$$

Take the 2-norm on both sides, get

$$\|f(x_2) - f(x_1)\| = \left\| \left( \int_0^1 \nabla f(x_1 + t(x_2 - x_1)) \, dt \right) (x_2 - x_1) \right\|$$
$$\leq \left\| \int_0^1 \nabla f(x_1 + t(x_2 - x_1)) \, dt \right\| \|x_2 - x_1\|$$
$$\leq \left( \int_0^1 \|\nabla f(x_1 + t(x_2 - x_1))\| \, dt \right) \|x_2 - x_1\|$$
$$\leq \max_{x' \in \mathcal{B}(x,\rho)} \|\nabla f(x')\| \cdot \|x_2 - x_1\|.$$

Therefore,

$$\max_{x' \in \mathcal{B}(x,\rho)} \|\nabla f(x')\| \geq \frac{\|f(x_1) - f(x_2)\|}{\|x_1 - x_2\|}, \ \forall x_1, x_2 \in \mathcal{B}(x,\rho),$$

which means

$$\max_{x' \in \mathcal{B}(x,\rho)} \|\nabla f(x')\| \geq \max_{x_1, x_2 \in \mathcal{B}(x,\rho)} \frac{\|f(x_1) - f(x_2)\|}{\|x_1 - x_2\|}$$
$$= K_x.$$

Considering both (a) and (b), we know that $\|\nabla f(x')\| \leq K_x, \forall x' \in \mathcal{B}(x,\rho)$ and $\max_{x' \in \mathcal{B}(x,\rho)} \|\nabla f(x')\| \geq K_x$. Therefore, we can conclude that $\max_{x' \in \mathcal{B}(x,\rho)} \|\nabla f(x')\| = K_x$. $\qquad\square$

**Theorem A.2** (Lipschitz's Jacobian Approximation). *Let $f : \mathbb{R}^n \to \mathbb{R}^m$ be a continuously differentiable neural network. The Jacobian norm $\|\nabla_x f\|$ is an approximation of $f$'s local Lipschitz constant within neighborhood $\mathcal{B}(x,\rho)$, centered at $x$ with radius $\rho$. The approximation error is*

$$\max_{\delta \in \mathcal{B}(0,\rho)} \left[ (\nabla_x \|\nabla_x f(x)\|)^\top \delta + o(\delta) \right],$$

*where $o(\delta)$ is a higher-order infinitesimal term with respect to $\delta$. Moreover, as $\rho \to 0$, the Jacobian norm converges to the exact local Lipschitz constant, i.e.*

$$\lim_{\rho \to 0} \|\nabla_x f\| = K(x).$$

*Proof.* By Definition 3.1 and Lemma A.1, we know that

$$K(x) = \max_{x_1, x_2 \in \mathcal{B}(x,\rho)} \frac{\|f(x_1) - f(x_2)\|}{\|x_1 - x_2\|}$$
$$= \max_{x' \in \mathcal{B}(x,\rho)} \|\nabla f(x')\|$$
$$= \max_{\delta \in \mathcal{B}(0,\rho)} \|\nabla f(x + \delta)\|.$$

By conducting the first-order Taylor expansion for $\|\nabla f(x + \delta)\|$, we get that

$$K(x) = \max_{\delta \in \mathcal{B}(0,\rho)} \left[ \|\nabla_x f(x)\| + (\nabla_x \|\nabla_x f(x)\|)^\top \delta + o(\delta) \right]$$
$$= \|\nabla_x f(x)\| + \max_{\delta \in \mathcal{B}(0,\rho)} \left[ (\nabla_x \|\nabla_x f(x)\|)^\top \delta + o(\delta) \right].$$

Therefore, the Jacobian norm $\|\nabla_x f\|$ is an approximation of the local Lipschitz constant $K(x)$, with the approximation error $\max_{\delta \in \mathcal{B}(0,\rho)} \left[ (\nabla_x \|\nabla_x f(x)\|)^\top \delta + o(\delta) \right]$. Furthermore, when $\rho \to 0$, the approximation error also approaches $0$. This means that $\|\nabla_x f\| \to K(x)$ when $\rho \to 0$. $\qquad\square$

## B. Fundamental Reasons of Action Fluctuation

**Non-smoothness of policy network.** A non-smooth policy network means that RL fits a non-smooth policy function mapping from the state to control action. The mapping function has significant output differences even if the inputs are closely adjacent. Consequently, when the state changes with time, a non-smooth action sequence is generated. Figure 14 visualizes the effect of policy non-smoothness.

**Existence of observation noise.** The noise results in the discontinuous changes in observations, making the actions produced by the policy network at the adjacent time stamps erratically differ. Even if the policy function fitted by the policy network is smooth enough, actions can still be fluctuated because of the erratic observation noise.

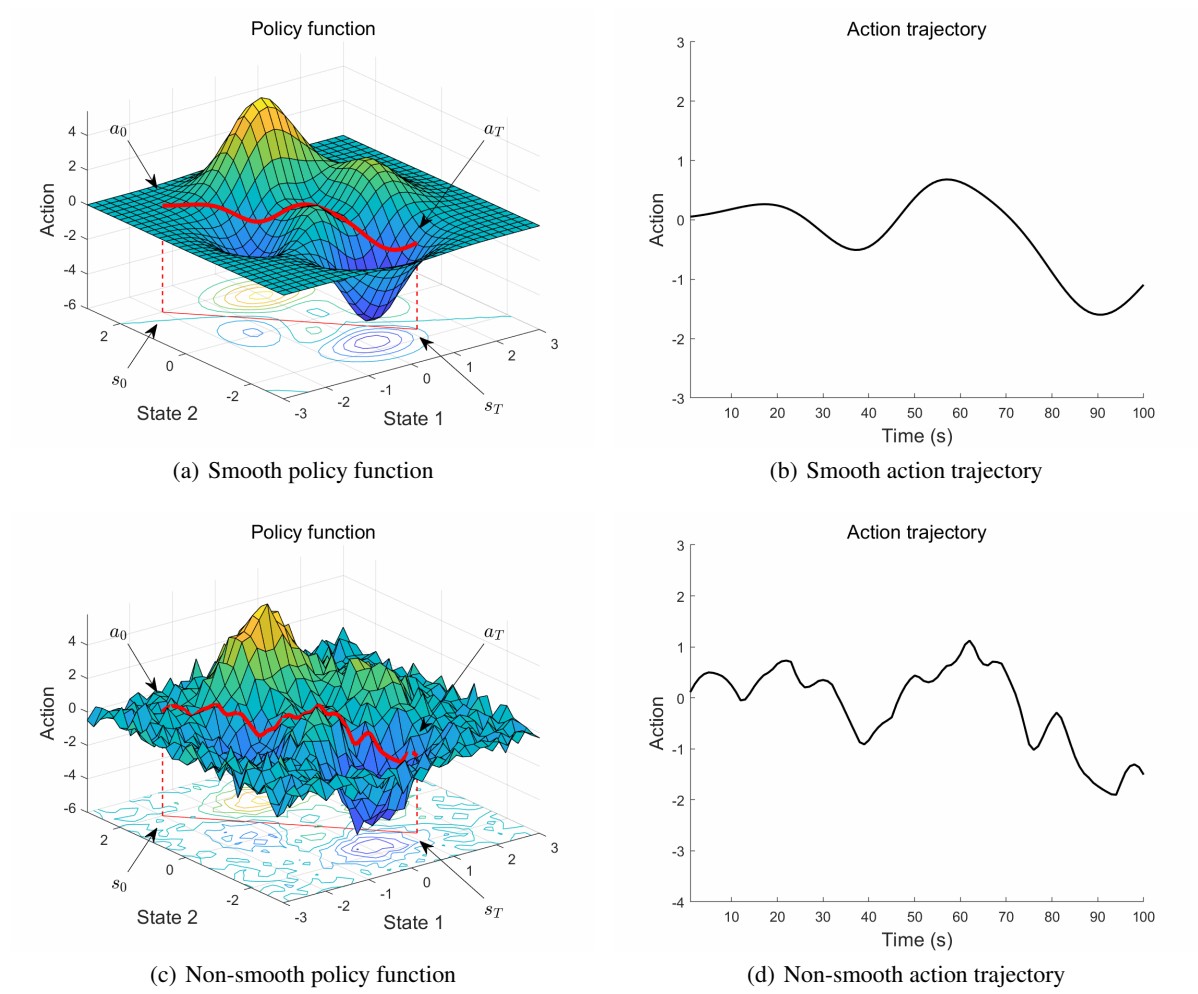

(a) Smooth policy function

(b) Smooth action trajectory

(c) Non-smooth policy function

(d) Non-smooth action trajectory

Figure 14: **Effect of policy non-smoothness.**

## C. Pseudocode of LipsNet++

---

**Algorithm 1** Forward and backward propagations of LipsNet++

---

1: **Input:** last $N$-step observations $o_t, o_{t-1}, \cdots, o_{t-N+1}$, original actor loss $\mathcal{L}$, original network parameter $\theta$.

    `/* Forward propagation */`
2: $x \leftarrow [o_t \, o_{t-1} \, \cdots \, o_{t-N+1}]^\top$
3: $X \leftarrow \text{FFT}(x)$
4: $\tilde{X} \leftarrow \text{symmetrize}(X_{\text{half}} \odot H)$
5: $\tilde{x} \leftarrow \text{IFFT}\left(\tilde{X}\right)$
6: $\tilde{o}_t \leftarrow$ the first row in $\tilde{x}$
7: $a_t \leftarrow f(\tilde{o}_t)$

    `/* Backward propagation */`
8: $\mathcal{L}'' \leftarrow \mathcal{L} + \lambda_k \, \|\nabla_{\tilde{o}_t} f\| + \lambda_h \, \|H\|_F$
9: $\theta_{\text{new}} \leftarrow \theta - \eta \nabla_\theta \mathcal{L}''$

10: **Output:** control action $a_t$, updated network parameter $\theta_{\text{new}}$.

---

# D. Double Integrator: Detailed Implementation and Results

The double integrator is a classic control task with linear dynamics and quadratic cost function, namely linear quadratic (LQ) control task. The environment used in this paper is a particle-moving environment. We train in noise-free environment and test in noisy environment with various noise level to comprehensively evaluate policy networks. We use a model-based RL algorithm, INFADP (Li, 2023), to train different policy networks including MLP (Rumelhart et al., 1986), MLP-SN (Takase et al., 2020), LipsNet-G (Song et al., 2023), LipsNet-L (Song et al., 2023), and LipsNet++. The hyperparameters for INFADP are listed in Table 2.

We set 5 different observation noise amplitudes and compare the performances of MLP, MLP-SN, LipsNet-G, LipsNet-L, and LipsNet++. Table 3 and Table 4 summarize the TAR and AFR, respectively. Figure 7 shows the variation trends of TAR and AFR as the noise increases. As shown in Figure 7(a), the TAR of LipsNet++ decreases much slower than that of the other networks. As shown in Figure 7(b), the AFR of LipsNet++ increases much slower than that of the other networks. These results indicate that LipsNet++ has superior action smoothness and noise robustness compared to previous works.

Table 2: **Hyperparameters for INFADP.**

| Hyperparameter | Value |
|---|---|
| Replay buffer capacity | 100000 |
| Buffer warm-up size | 1000 |
| Batch size | 64 |
| Discount $\gamma$ | 0.99 |
| Target network soft-update rate $\tau$ | 0.2 |
| Initial random interaction steps | 0 |
| Interaction steps per iteration | 8 |
| Network update times per iteration | 1 |
| Prediction step | 1 |
| Action bound | $[-5, 5]$ |
| Exploration noise std. deviation | 0 |
| Hidden layers in subnetwork $f$ | $[64, 64]$ |
| Activations in subnetwork $f$ | ReLU |
| Hidden layers in critic network | $[64, 64]$ |
| Activations in critic network | ReLU |
| Optimizer | Adam |
| Actor learning rate | $3 \cdot 10^{-4}$ |
| Critic learning rate | $8 \cdot 10^{-4}$ |
| length of historical obsv. $N$ | 8 |
| coefficient $\lambda_k$ | 0.01 |
| coefficient $\lambda_h$ | 1 |

Table 3: **Comparison of TAR on double integrator environment.** The observation noise in each dimension is distributed in $U(-\sigma, \sigma)$. The data in this table is visualized in Figure 7(a).

| Noise | Methods | | | | | | |
|---|---|---|---|---|---|---|---|
| $\sigma$ | MLP | CAPS | L2C2 | MLP-SN | LipsNet-G | LipsNet-L | LipsNet++ |
| 0.01 | **-51.0** $_{\pm 0.1}$ | -52.1 $_{\pm 0.1}$ | -55.0 $_{\pm 0.1}$ | -62.0 $_{\pm 0.1}$ | -53.2 $_{\pm 0.1}$ | -55.3 $_{\pm 0.1}$ | -56.5 $_{\pm 0.1}$ |
| 0.05 | -53.5 $_{\pm 0.2}$ | **-53.0** $_{\pm 0.2}$ | -55.4 $_{\pm 0.4}$ | -62.3 $_{\pm 0.4}$ | -54.3 $_{\pm 0.3}$ | -55.6 $_{\pm 0.4}$ | -56.8 $_{\pm 0.2}$ |
| 0.1 | -59.5 $_{\pm 0.6}$ | -56.9 $_{\pm 0.6}$ | -57.5 $_{\pm 0.5}$ | -62.8 $_{\pm 0.7}$ | **-54.2** $_{\pm 0.7}$ | -56.0 $_{\pm 0.6}$ | -57.1 $_{\pm 0.6}$ |
| 0.2 | -78.4 $_{\pm 1.8}$ | -67.9 $_{\pm 1.0}$ | -63.4 $_{\pm 1.5}$ | -65.9 $_{\pm 1.7}$ | -59.8 $_{\pm 1.1}$ | -58.7 $_{\pm 1.4}$ | **-57.9** $_{\pm 0.6}$ |
| 0.3 | -103.2 $_{\pm 3.7}$ | -85.9 $_{\pm 3.0}$ | -82.3 $_{\pm 4.4}$ | -71.8 $_{\pm 2.3}$ | -74.3 $_{\pm 2.1}$ | -65.3 $_{\pm 1.6}$ | **-59.9** $_{\pm 1.9}$ |

Table 4: **Comparison of AFR on double integrator environment.** The observation noise in each dimension is distributed in $U(-\sigma, \sigma)$. The data in this table is visualized in Figure 7(b).

| Noise | Methods | | | | | | |
|---|---|---|---|---|---|---|---|
| $\sigma$ | MLP | CAPS | L2C2 | MLP-SN | LipsNet-G | LipsNet-L | LipsNet++ |
| 0.01 | 0.02 ± 0.01 | 0.01 ± 0.01 | 0.01 ± 0.01 | 0.01 ± 0.01 | 0.01 ± 0.01 | 0.01 ± 0.01 | **0.00** ± 0.01 |
| 0.05 | 0.11 ± 0.01 | 0.07 ± 0.01 | 0.05 ± 0.01 | 0.03 ± 0.01 | 0.04 ± 0.01 | 0.03 ± 0.01 | **0.01** ± 0.01 |
| 0.1 | 0.19 ± 0.01 | 0.15 ± 0.01 | 0.10 ± 0.01 | 0.06 ± 0.01 | 0.07 ± 0.01 | 0.06 ± 0.01 | **0.02** ± 0.01 |
| 0.2 | 0.34 ± 0.02 | 0.26 ± 0.01 | 0.20 ± 0.01 | 0.13 ± 0.01 | 0.17 ± 0.01 | 0.12 ± 0.01 | **0.03** ± 0.01 |
| 0.3 | 0.48 ± 0.02 | 0.37 ± 0.02 | 0.33 ± 0.02 | 0.20 ± 0.01 | 0.28 ± 0.01 | 0.20 ± 0.01 | **0.05** ± 0.01 |

## E. Ablation Study for Two Techniques

In this appendix, we implement ablation study for the two techniques proposed in our paper, i.e. Jacobian regularization and Fourier filter layer. The two techniques respectively tackle the two fundamental reasons of action fluctuation, as described in Section 3.1. The Jacobian regularization enhances the smoothness of policy network by introducing the Jacobian norm in actor loss function. Similarly, the Fourier filter layer enhance the noise robustness of policy network by introducing the Frobenius norm of the filter matrix in actor loss function. The resulted actor loss is illustrated in Equation 4:

$$\mathcal{L}'' = \mathcal{L} + \lambda_k \left\| \nabla f \right\| + \lambda_h \left\| H \right\|_F .$$

In order to validate the effectiveness of each technique, the two coefficients $\lambda_k$ and $\lambda_h$ are set to zero in turn. The performance result on double integrator environment is shown in Figure 15. The performance result on DMControl's Cheetah and Walker environments is shown in Table 5. These results shows that setting either coefficient to zero will lead to a rapid decrease in TAR and a rapid increase in AFR when the noise increases. It indicates that both the Jacobian regularization and Fourier filter layer are effective techniques and they are both indispensable.

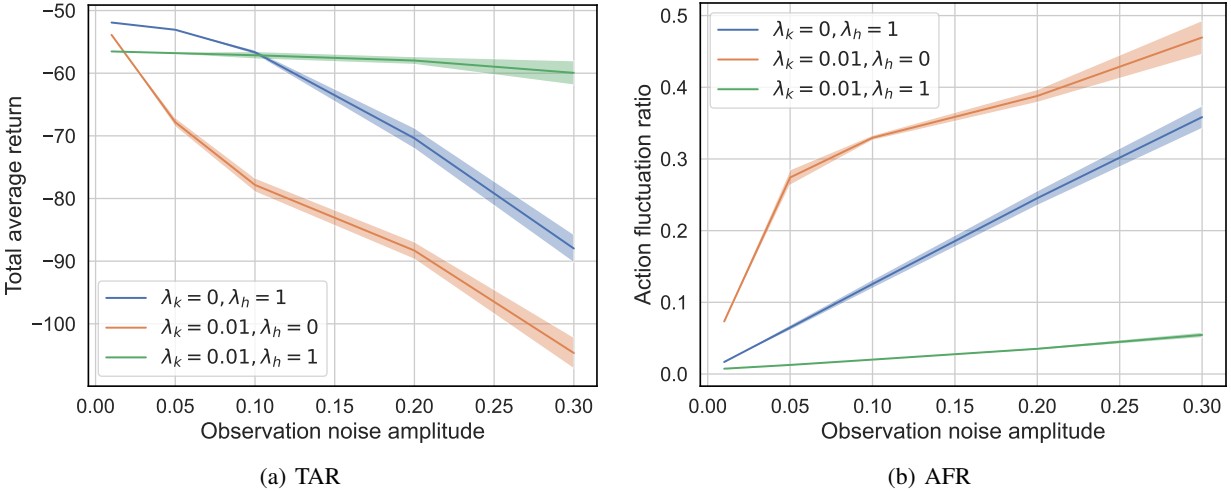

(a) TAR        (b) AFR

Figure 15: **Ablation study for Jacobian regularization and Fourier filter layer on double integrator environment.**

## F. Sensitivity Analysis for $\lambda_k$ and $\lambda_h$

In this appendix, we provide the sensitivity analysis for the hyperparameters $\lambda_k$ and $\lambda_h$. We design experiments to demonstrate that the two hyperparameters have low sensitivity, making LipsNet++ convenient for tuning and easy to use.

Similar to the approach in Appendix E, we fix one hyperparameter and then vary the other to observe the changes in TAR and AFR on double integrator environment. As shown in Figure 16, when $\lambda_h$ is fixed at 1 and $\lambda_k$ varies between 0.001,

Table 5: **Ablation study on Cheetah and Walker.** The result shows that setting either coefficient to zero will lead to an increase in AFR, which indicates the two techniques are all effective and indispensable.

| Environment | $\lambda_k$ | $\lambda_h$ | Total average return | Action fluctuation ratio |
|---|---|---|---|---|
| Cheetah | $10^{-3}$ | $10^{-3}$ | **822** $_{\pm 11}$ | **0.94** $_{\pm 0.01}$ |
| | 0 | $10^{-3}$ | 822 $_{\pm 15}$ | 1.08 $_{\pm 0.02}$ |
| | $10^{-3}$ | 0 | 821 $_{\pm 17}$ | 1.21 $_{\pm 0.02}$ |
| Walker | $10^{-2}$ | $10^{-3}$ | **961** $_{\pm 12}$ | **0.78** $_{\pm 0.01}$ |
| | 0 | $10^{-3}$ | 958 $_{\pm 15}$ | 0.98 $_{\pm 0.02}$ |
| | $10^{-2}$ | 0 | 940 $_{\pm 14}$ | 1.89 $_{\pm 0.01}$ |

0.01, and 0.1, the performance differences are significant. However, when $\lambda_h$ is fixed at 1 and $\lambda_k$ varies between 0.01 and 0.02, the performances are essentially consistent. A similar phenomenon can also be found for hyperparameter $\lambda_h$, as shown in Figure 17. When $\lambda_k$ is fixed at 0.01 and $\lambda_h$ varies between 0.1, 1 and 10, the performance differences are significant. However, when $\lambda_h$ is fixed at 1 and $\lambda_k$ varies between 1 and 2, the performances are essentially consistent.

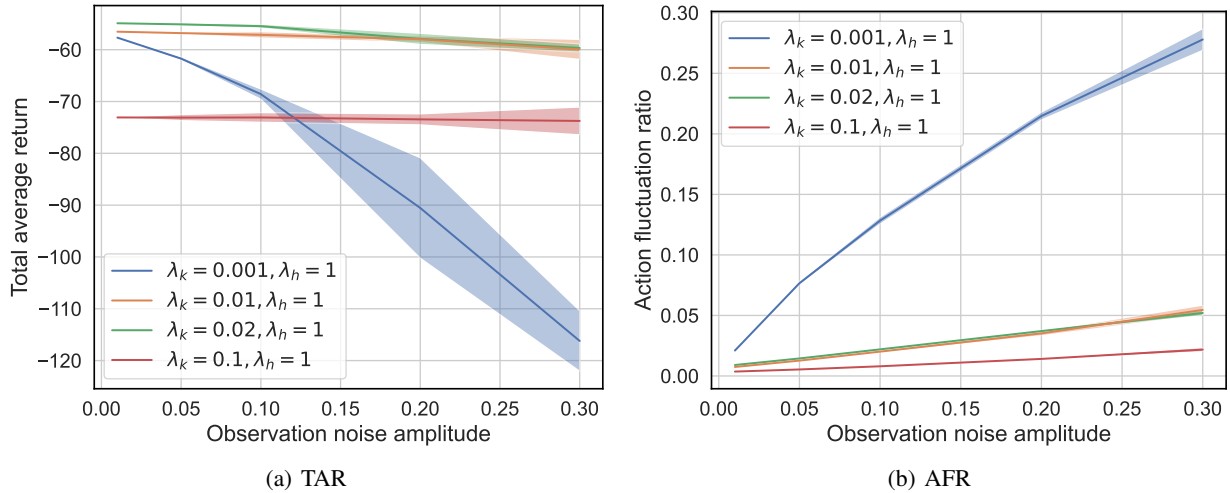

(a) TAR        (b) AFR

Figure 16: **Sensitivity analysis for $\lambda_k$.**

The above results imply that the hyperparameter $\lambda_k$ and $\lambda_h$ have low sensitivity. Only the magnitude of hyperparameters have a significant impact on performance, while changing the hyperparameters within an appropriate magnitude has a minimal effect on performance. Therefore, when tuning parameters, the user only need to set an appropriate magnitude. It makes LipsNet++ convenient for tuning and easy to use.

## G. Sensitivity Analysis for $N$

In this appendix, we provide the sensitivity analysis for hyperparameter $N$, which represents the length of historical observations. We design experiments to demonstrate that $N$ exhibits low sensitivity when the length of historical observations is sufficiently long, making LipsNet++ convenient for tuning and easy to use.

The values of $N$ are set to range from 1 to 32 in the double integrator environment. The Figure 18(a) and (b) show the trend of TAR and AFR when $N$ changes. The result shows that the performance no longer improves once $N$ exceeds a threshold, suggesting a low sensitivity. Therefore, users only need to set a relatively large value of $N$, making LipsNet++ very convenient for tuning.

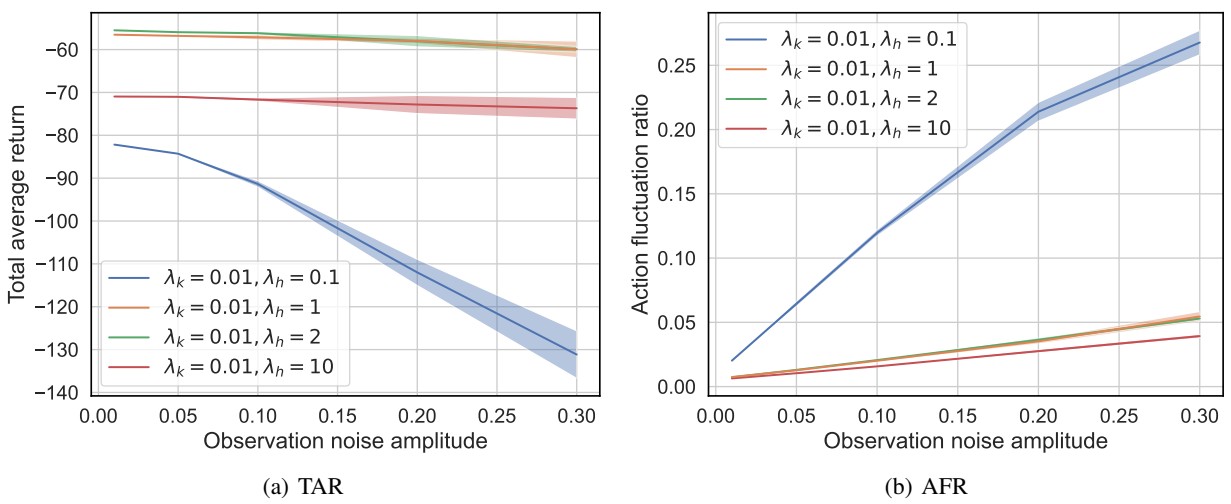

(a) TAR

(b) AFR

Figure 17: **Sensitivity analysis for** $\lambda_h$**.**

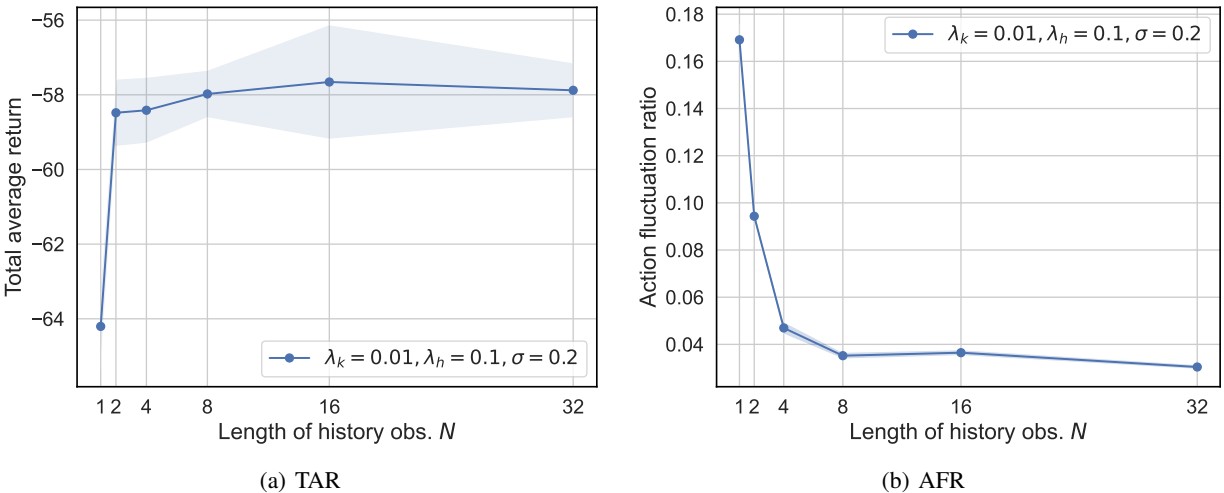

(a) TAR

(b) AFR

Figure 18: **Sensitivity analysis for** $N$**.**

## H. Computational Efficiency Analysis

To evaluate the computational efficiency, we provide the detailed forward and backward processing time of policy networks. All policy networks are from the network trained in double integrator environment. This analysis is implemented on AMD Ryzen Threadripper 3960X 24-Core Processor. For MLP-SN, the number of power iterations is set to 1, whose time usage is included in the backward stage. Similarly, the computation times for Jacobian norm and Frobenius norm in LipsNet++ are included in the backward stage. The length of historical observations used in LipsNet++ is 8.

The results are summarized in Table 6. Compared to the previous SOTA network LipsNet, LipsNet++ has significantly faster speed for forward propagation. This allows LipsNet++ to be applied in high-real-time tasks. We acknowledge that backward propagation speed of LipsNet++ is relatively slow, but we have devised a solution to accelerate this in future work by using multiple forward propagation and zero-order gradient estimation to compute the Jacobian matrix.

Table 6: Forward and backward **propagation time comparison**.

| Settings | | Policy network | | | |
|----------|------------|---------|---------|-----------|-----------|
| Propagation | Batch size | MLP | MLP-SN | LipsNet-L | LipsNet++ |
| forward | 1 | 0.10 ms | 0.11 ms | 0.75 ms | 0.16 ms |
| | 100 | 0.11 ms | 0.12 ms | 1.41 ms | 0.25 ms |
| backward | 1 | 0.17 ms | 0.76 ms | 0.45 ms | 1.98 ms |
| | 100 | 0.28 ms | 0.89 ms | 0.73 ms | 2.48 ms |

Since network propagation times constitute only a part of the overall RL training process, we next compare the total training wall-clock times. Table 7 presents the wall-clock times for 1M iterations of TD3 on the DMControl environments. The results show that, on average, the training time of LipsNet++ is 1.6 times that of MLP. The difference in wall-clock time is not as significant as the difference in backward time shown in Table 6, as RL algorithms involve additional time-consuming steps beyond backward, such as sampling and evaluation.

Table 7: **Training wall-clock time comparison.** The data show the wall-clock times used for 1M iterations in TD3. On average, the training time of LipsNet++ is 1.6 times that of MLP.

| Network | Env | | | | Total |
|---------|---------|---------|---------|---------|-------|
| | Cartpole | Reacher | Cheetah | Walker | |
| MLP | 120 min | 118 min | 128 min | 125 min | 491 min |
| LipsNet++ | 194 min | 195 min | 206 min | 204 min | 799 min |

In conclusion, LipsNet++'s forward time is under 0.2 ms, making it suitable for real-time applications. While the training wall-clock time shows a slight increase, it remains acceptable and has clear pathways for future optimization.

## I. DeepMind Control Suite Benchmark

The DeepMind Control Suite (DMControl) (Tassa et al., 2018) encompasses a collection of meticulously crafted continuous control tasks. These environments feature consistent structures, rewards that are both interpretable and normalized, facilitating a more straightforward comparison of performance across different algorithms. Developed in Python and leveraging the MuJoCo physics engine (Todorov et al., 2012), DMControl currently stands as one of the most esteemed benchmarks for evaluating RL and continuous control tasks.

In DMControl, the term "domain" denotes a specific physical model, whereas a "task" corresponds to an instantiation of that model with a defined Markov Decision Process (MDP) structure. For instance, within the cartpole domain, the distinction between the `swingup` and `balance` tasks lies in the initial orientation of the pole: it is initialized pointing downward in the `swingup` task and upward in the `balance` task, respectively. In the following figures, we provide detailed descriptions of the domains used in this paper, with each domain's name followed by a tuple of three integers that denote the dimensions of the state, action, and observation spaces, respectively, formatted as $\left( \dim\left(\mathcal{S}\right), \dim\left(\mathcal{A}\right), \dim\left(\mathcal{O}\right) \right)$.

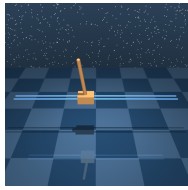

Figure 19: **Cartpole**(4, 1, 5): This domain features a cart connected to a pole via an unactuated joint. It encompasses a set of four distinct tasks. In the context of our experimental setup, we focus on the `swingup` task. Here, the pole is initially positioned downward, and the objective is to apply appropriate forces to the cart to swing the pole upward and maintain its upright position.

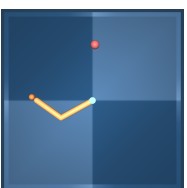

Figure 20: **Reacher**(4, 2, 7): This domain comprises two interconnected poles with a sphere whose initial position is randomly determined. One end of the linked poles is anchored at the origin of the coordinate space, while the other remains free to move. The domain offers two distinct tasks, and we focus on the `easy` task. The task requires the application of forces to the pendulum to ensure that its endpoint remains within the red area at all times.

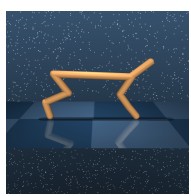

Figure 21: **Cheetah**(18, 6, 17): This domain features a planar bipedal and it is able to crawl forward by its two legs. It involves a single task, namely the `run` task. In the initial state of the environment, the agent's pose is random, typically in a non-standing position. In this task, the challenge is to control the planar biped to achieve an upright standing position and subsequently propel it forward into a running motion with a targeted forward velocity.

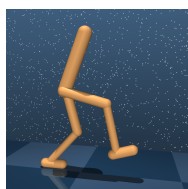

Figure 22: **Walker**(18, 6, 24): This domain includes a planar walker. This environment simulates a simple locomotion task of humans, with the agent possessing two legs and advancing in an upright posture. It comprises three distinct tasks, and our experiment focus the `walk` task. In this task, the objective is to control the walker to maintain an upright torso posture, achieve the specified torso height, and maintain a consistent forward velocity.

## J. DMControl: Detailed Implementation and Results

We employ the Twin Delayed Deep Deterministic Policy Gradient (TD3) (Fujimoto et al., 2018), a model-free RL algorithm, to train on DMControl. The hyperparameters for TD3 remain consistent across all environments, except for the coefficients $\lambda_k$, $\lambda_h$, and the length of historical observations $N$. The hyperparameters for TD3 are listed in Table 8. The environment-related hyperparameters are listed in Table 9.

To evaluate comprehensively, networks are tested on both noise-free and noisy environments. For noisy environments, the noise amplitudes are listed in Table 10. We compare LipsNet++ with MLP, LipsNet-G, LipsNet-L using 10 seeds. All results are summarized in Table 11 and 12, from which we can find that LipsNet++ has the highest TAR and the lowest AFR in all cases. These results imply LipsNet++ has good action smoothness and noise robustness.

For comparing LipsNet++ and MLP-SN, we train them on DMControl Reacher environment. We use a 3-layer MLP-SN network and manually tuning its spectral norm of each layer by grid search. The global Lipschitz constant of MLP-SN is the product of the spectral norms of all layers. The results are listed in Table 13, from which we can find that LipsNet++ outperforms MLP-SN under all hyperparameter settings. As shown in Figure 23, the trends of TAR and AFR imply that even with fine adjustments to the spectral norm, the overall performance of MLP-SN remains inferior to that of LipsNet++. We refrain from comparing LipsNet++ to MLP-SN across all environments used in this paper, because this would necessitate the manual tuning of spectral norm hyperparameters for each layer, which have an unwieldy number of potential hyperparameter combinations.

Additionally, the learned filter matrix $H$ in LipsNet++ is visualized in Figure 24 to show the noise filtering ability. Figure 24(a) and 24(b) show the frequency distributions of observation in noise-free environment and noisy environment, respectively. Their shades of color represent the intensity of frequency. The color in Figure 24(c) denotes the magnitude of elements in matrix $H$, which determines which frequencies are suppressed or strengthened. The result implies that

Table 8: **Hyperparameters for TD3.**

| Hyperparameter | Value |
|---|---|
| Replay buffer capacity | 1000000 |
| Buffer warm-up size | 1000 |
| Batch size | 100 |
| Discount $\gamma$ | 0.99 |
| Target network soft-update rate $\tau$ | 0.005 |
| Target noise | 0.2 |
| Target noise limit | 0.5 |
| Exploration noise std. deviation | 0.1 |
| Policy delay times | 2 |
| Initial random interaction steps | 25000 |
| Interaction steps per iteration | 50 |
| Network update times per iteration | 50 |
| Hidden layers in subnetwork $f$ | $[64, 64]$ |
| Activations in subnetwork $f$ | ReLU |
| Hidden layers in critic network | $[64, 64]$ |
| Activations in critic network | ReLU |
| Optimizer | Adam |
| Actor learning rate | $1 \cdot 10^{-3}$ |
| Critic learning rate | $1 \cdot 10^{-3}$ |

Table 9: **Environment-related hyperparameters in DMControl.**

| Env | $\lambda_k$ | $\lambda_h$ | Length of his. obsv. $N$ |
|---|---|---|---|
| Cartpole | $10^{-2}$ | $10^{-2}$ | 5 |
| Reacher | $10^{-2}$ | $10^{-3}$ | 5 |
| Cheetah | $10^{-3}$ | $10^{-3}$ | 5 |
| Walker | $10^{-2}$ | $10^{-3}$ | 10 |

Table 10: **Observation noise in DMControl.** The observation noise in each dimension is distributed in $U(-\sigma, \sigma)$.

| Env | Noise amplitude $\sigma$ |
|---|---|
| Cartpole | $[0.1, 0.1, 0.1, 0.2, 0.2]$ |
| Reacher | $[0.001 \text{ repeats } 7 \text{ times}]$ |
| Cheetah | $[0.01, 0.01, 0.05, 0.05, 0.05, 0.05, 0.05, 0.05,$ $0.5, 0.05, 0.1, 0.5, 0.5, 0.5, 0.5, 0.5, 0.5]$ |
| Walker | $[0.25 \text{ repeats } 24 \text{ times}]$ |

Table 11: **Total average return in DMControl.**

| Environment | | MLP | LipsNet-G | LipsNet-L | LipsNet++ |
|---|---|---|---|---|---|
| noise-free env | Cartpole | $805 \pm 0.8$ | $691 \pm 1.0$ | $831 \pm 0.9$ | $\mathbf{841} \pm 0.2$ |
| | Reacher | $981 \pm 10$ | $979 \pm 11$ | $983 \pm 10$ | $\mathbf{988} \pm 10$ |
| | Cheetah | $816 \pm 30$ | $702 \pm 10$ | $822 \pm 4$ | $\mathbf{829} \pm 15$ |
| | Walker | $926 \pm 12$ | $956 \pm 20$ | $945 \pm 13$ | $\mathbf{962} \pm 10$ |
| noisy env | Cartpole | $763 \pm 9$ | $517 \pm 41$ | $823 \pm 6$ | $\mathbf{825} \pm 3$ |
| | Reacher | $972 \pm 25$ | $973 \pm 18$ | $978 \pm 17$ | $\mathbf{982} \pm 10$ |
| | Cheetah | $813 \pm 29$ | $680 \pm 7$ | $818 \pm 11$ | $\mathbf{822} \pm 11$ |
| | Walker | $911 \pm 26$ | $942 \pm 15$ | $929 \pm 11$ | $\mathbf{961} \pm 12$ |

Table 12: **Action fluctuation ratio in DMControl.**

| Environment | | MLP | LipsNet-G | LipsNet-L | LipsNet++ |
|---|---|---|---|---|---|
| noise-free env | Cartpole | $0.04 \pm 0.00$ | $0.08 \pm 0.00$ | $0.01 \pm 0.00$ | $\mathbf{0.01} \pm 0.00$ |
| | Reacher | $2.07 \pm 0.60$ | $0.13 \pm 0.24$ | $0.01 \pm 0.00$ | $\mathbf{0.01} \pm 0.00$ |
| | Cheetah | $1.08 \pm 0.02$ | $0.92 \pm 0.01$ | $0.94 \pm 0.01$ | $\mathbf{0.90} \pm 0.01$ |
| | Walker | $1.89 \pm 0.02$ | $1.25 \pm 0.02$ | $0.93 \pm 0.01$ | $\mathbf{0.74} \pm 0.01$ |
| noisy env | Cartpole | $0.58 \pm 0.03$ | $0.75 \pm 0.09$ | $0.17 \pm 0.01$ | $\mathbf{0.13} \pm 0.00$ |
| | Reacher | $2.41 \pm 0.28$ | $0.04 \pm 0.00$ | $0.04 \pm 0.03$ | $\mathbf{0.01} \pm 0.00$ |
| | Cheetah | $1.13 \pm 0.02$ | $1.00 \pm 0.01$ | $1.08 \pm 0.01$ | $\mathbf{0.94} \pm 0.01$ |
| | Walker | $2.02 \pm 0.03$ | $1.68 \pm 0.01$ | $1.21 \pm 0.01$ | $\mathbf{0.78} \pm 0.01$ |

Table 13: **Performance of LipsNet++ and MLP-SN on DMControl Reacher.**

| Network | | Total average return | Action fluctuation ratio |
|---|---|---|---|
| Name | Spectral norm for each layer | | |
| MLP-SN | 5.0 | $760 \pm 381$ | $0.01 \pm 0.00$ |
| | 5.5 | $831 \pm 102$ | $0.01 \pm 0.00$ |
| | 5.8 | $954 \pm 10$ | $0.08 \pm 0.05$ |
| | 6.0 | $967 \pm 28$ | $0.13 \pm 0.08$ |
| LipsNet++ | | $\mathbf{988} \pm 10$ | $\mathbf{0.01} \pm 0.00$ |

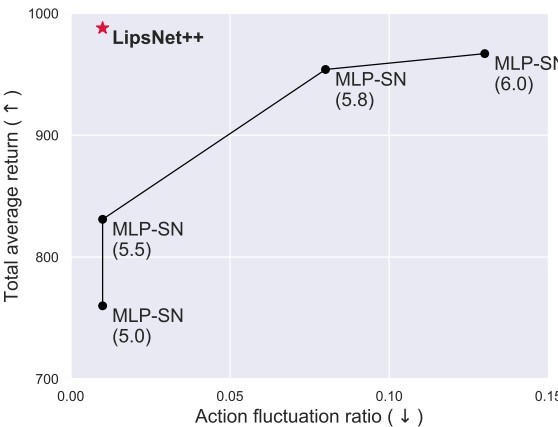

Figure 23: **Performance comparison of LipsNet++ and MLP-SN** on DMControl Reacher. The numbers in parentheses are the spectral norm of each layer in MLP-SN. Points that closer to the top-left corner indicate better performance. It shows that even with fine adjustments to spectral norm, MLP-SN's overall performance remains inferior to that of LipsNet++.

the learned filter matrix mainly focus on the frequencies that containing observation information, and rarely focus on the frequencies that containing noises. In other words, LipsNet++ can automatically extract the important frequencies and filter out the noise frequencies.

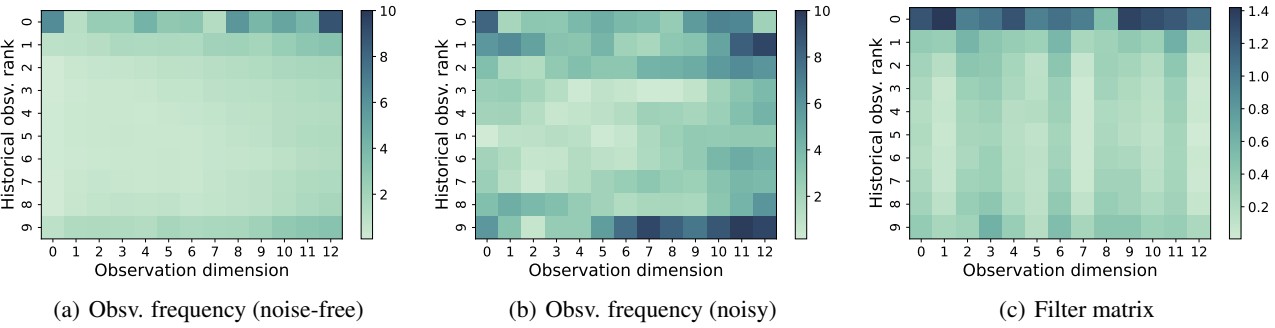

(a) Obsv. frequency (noise-free)       (b) Obsv. frequency (noisy)       (c) Filter matrix

Figure 24: **Filter matrix and observation frequency in walker environment.** The color in (a) and (b) represents the intensity of frequency. The color in (c) represents the magnitude of elements in matrix $H$. The color distribution in (c) implies LipsNet++ can automatically extract the important frequencies and filter out the noise frequencies.

## K. Comparison to Reward Penalty

Punishing the difference between consecutive actions in the reward is an effective way to smooth the actions in some environments. However, such an approach breaks the Markov property, which affects the performance, albeit to a minor extent in certain environments. Moreover, we found that adding reward penalty in a sparse reward environment increases action fluctuation rather than smoothing it, which is consistent with the finding by Chen et al. (2021) and Song et al. (2023).

Cartpole in DMControl is a sparse reward environment. The reward is 1 when the pole is within $30°$ of the vertical and 0 otherwise. We implement TD3 in this environment, punishing the difference between consecutive actions in the reward. Specifically, the new reward is $r = r_{\mathrm{origin}} + \alpha \|a_{t+1} - a_t\|$, where $r_{\mathrm{origin}}$ is the original sparse reward, $\alpha$ is the penalty coefficient and $a_{t+1}$ is the output of actor network under $s_{t+1}$. The experiment results are summarized in Table 14. The results imply that simply adding reward penalty in the sparse reward environment increases the action fluctuation ratio. Superiorly, LipsNet++ can smooth actions even in the sparse reward environment.

Table 14: Comparison to reward penalty.

| Method | Penalty coefficient $\alpha$ | Total average return | Action fluctuation ratio |
|---|---|---|---|
| TD3 (MLP, reward penalty) | 0.01 | $825 \pm 0.5$ | $0.27 \pm 0.01$ |
| TD3 (MLP, reward penalty) | 0.1 | $819 \pm 0.8$ | $0.21 \pm 0.01$ |
| TD3 (MLP, reward penalty) | 1 | $13 \pm 0.5$ | $0.02 \pm 0.00$ |
| TD3 (MLP) | | $805 \pm 0.8$ | $0.04 \pm 0.00$ |
| TD3 (LipsNet-G) | | $691 \pm 1.0$ | $0.08 \pm 0.00$ |
| TD3 (LipsNet-L) | | $831 \pm 0.9$ | $0.01 \pm 0.00$ |
| TD3 (LipsNet++) | | $\mathbf{841} \pm 0.2$ | $\mathbf{0.01} \pm 0.00$ |

## L. Mini-Vehicle Driving: Introduction of Vehicle and Task

The vehicle robot is driven by two differential wheels, which is shown in Figure 26. The task for the robot is to track a given reference trajectory and reference velocity while avoiding obstacle. The setting of observations and actions in this environment is described in Table 15.

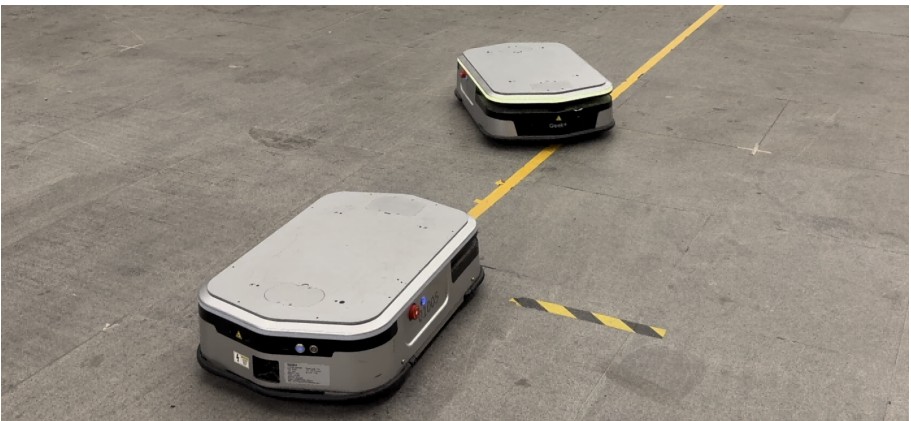

Figure 25: **Physical vehicle robots.**

For the perception, the vehicle is equipped with LiDAR, obtaining its position by matching with a pre-scanned point cloud map generated by SLAM. In this way, vehicle can detect its horizontal coordinate $x$, vertical coordinate $y$, and heading angle $\phi$. The vehicle is also equipped with a speed sensor that measures the linear velocity $v$ and angular velocity $\omega$. To increase the complexity of the task, another vehicle is used as a obstacle vehicle. Both vehicles can exchange real-time state information with each other via WiFi communication.

For the decision-making and control, a policy network trained by RL is deployed on the vehicle. After inputting the perceived observation into the network, control actions are computed, namely linear acceleration $\dot{v}$ and angular acceleration $\dot{\omega}$. Then, control actions are sent to the motor to execute the command. The overall control mode is shown in Figure 27.

As illustrated in Section 4.3, there are four diverse scenarios in this environment. The scenario descriptions are listed in Table 1. To describe the scenario settings more clearly, Figure 28 shows the map and vehicle routes for each scenario. Figure 29 shows the corresponding snapshot for each scenario. In scenarios 1-3, the obstacle vehicle goes straight with constant speed. In scenario 4, the obstacle vehicle is manipulated by human.

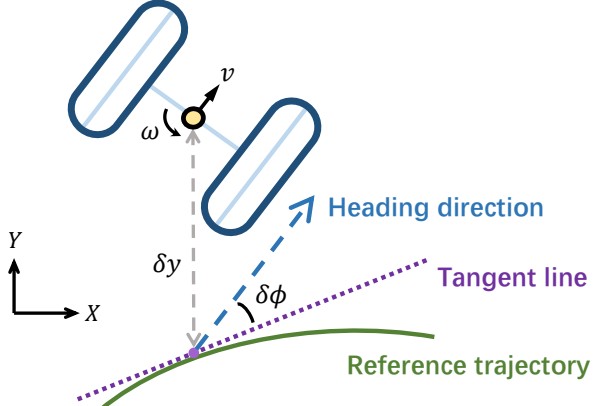

Figure 26: **Vehicle kinematics model.** The vehicle moves by two differential wheels, tracking the reference trajectory.

Table 15: **Variables in mini-vehicle driving env.**

| Variable | | Description |
|---|---|---|
| Obsv. | $v$ | longitudinal speed |
| | $\omega$ | yaw rate |
| | $\delta y$ | trajectory offset |
| | $\delta \phi$ | heading angle error |
| | $\delta v$ | speed error |
| | $\Delta x$ | obstacle's relative $x$ position |
| | $\Delta y$ | obstacle's relative $y$ position |
| | $\Delta \phi$ | obstacle's relative angle |
| | $\Delta v$ | obstacle's relative speed |
| | $\Delta \omega$ | obstacle's relative yaw rate |
| Action | $\dot{v}$ | longitudinal acceleration |
| | $\dot{\omega}$ | yaw acceleration |

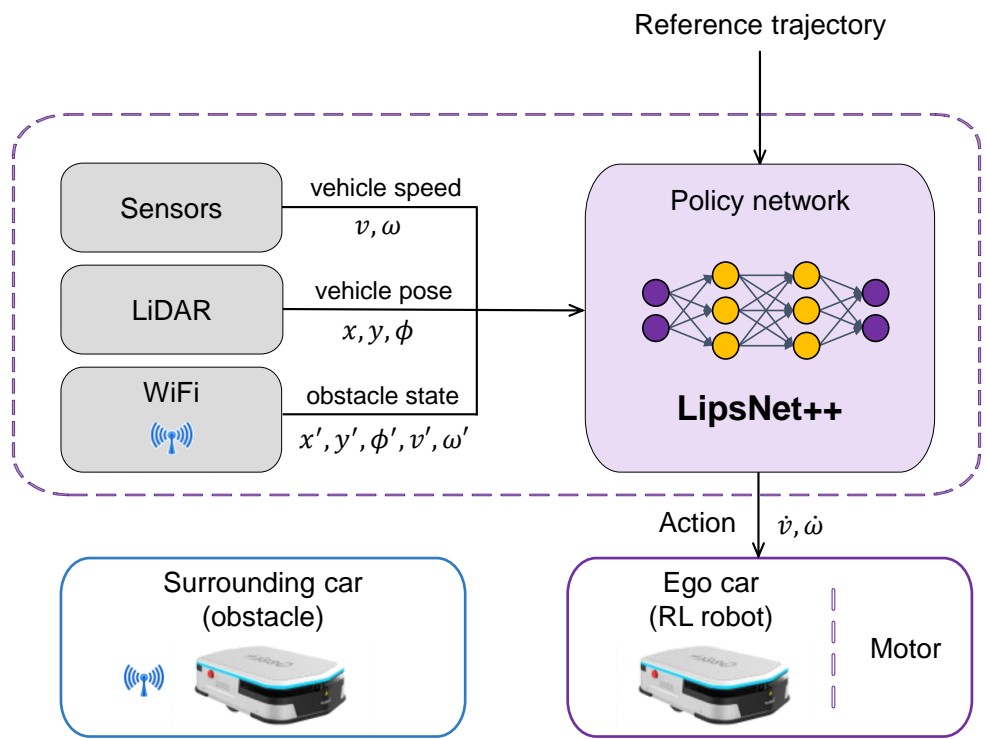

Figure 27: **Flowchart of vehicle control mode.**

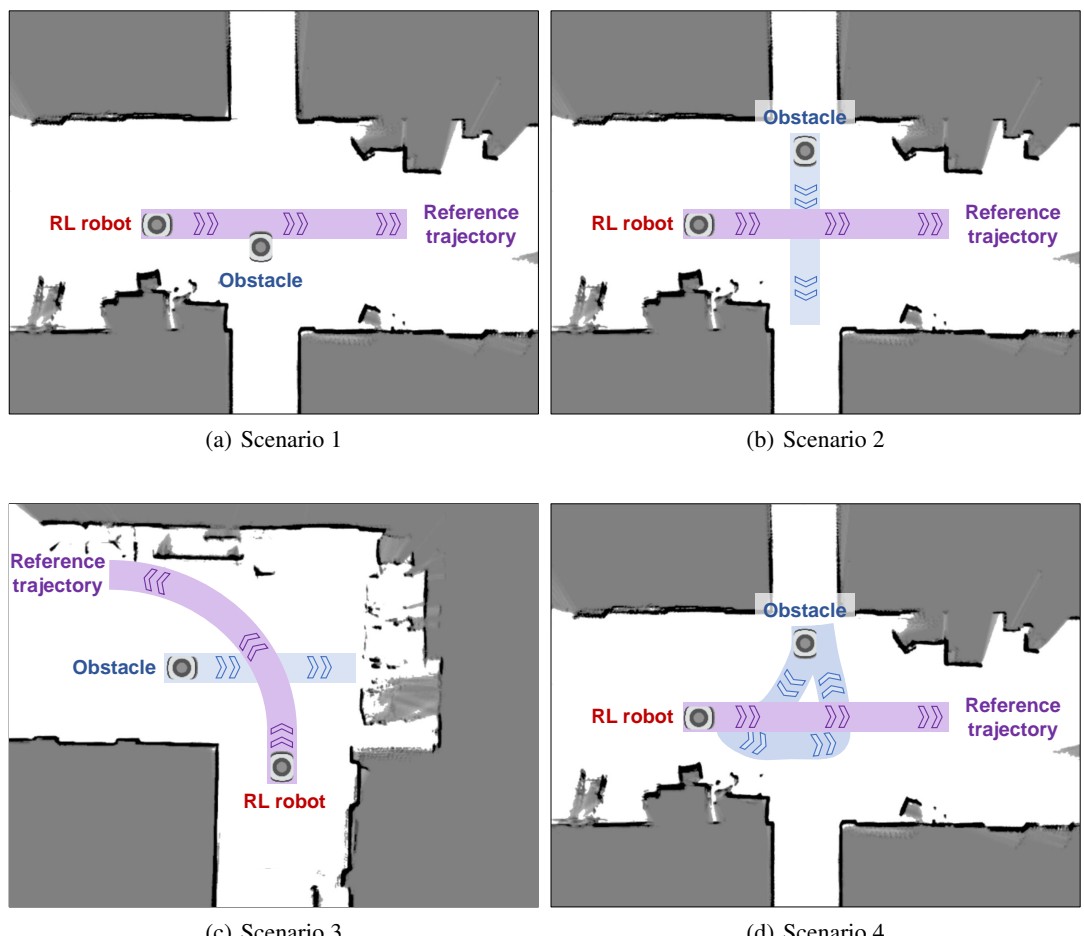

(a) Scenario 1          (b) Scenario 2

(c) Scenario 3          (d) Scenario 4

Figure 28: **Scenario illustration of mini-vehicle driving environment.**

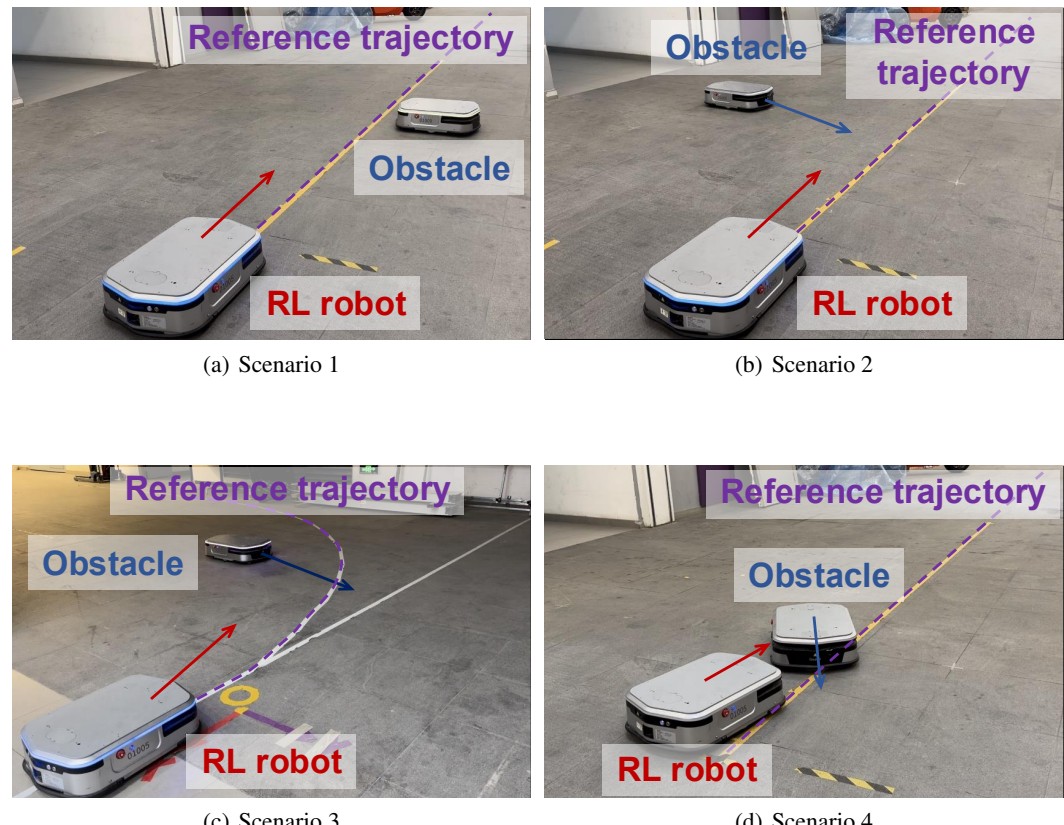

(a) Scenario 1

(b) Scenario 2

(c) Scenario 3

(d) Scenario 4

Figure 29: **Scenario snapshots of mini-vehicle driving environment.**

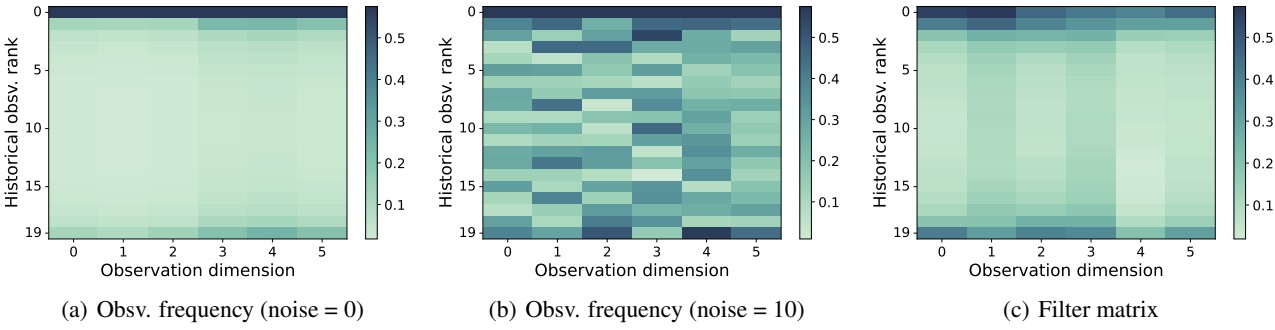

(a) Obsv. frequency (noise = 0)

(b) Obsv. frequency (noise = 10)

(c) Filter matrix

Figure 30: **Filter matrix and observation frequency in mini-vehicle driving environment.** The color in (a) and (b) represents the intensity of frequency. The color in (c) represents the magnitude of elements in matrix $H$. The color distribution in (c) implies LipsNet++ can automatically extract the important frequencies and filter out the noise frequencies.

## M. Mini-Vehicle Driving: Detailed Implementation and Results

In the training stage, observation noise is set to zero. In the vehicle testing stage, multiple different magnitudes of observation noise are added to thoroughly test the performance of policy networks. The noise magnitude is adjusted using the coefficient $\sigma_{\text{coef}} \in \mathbb{R}^+ \cup \{0\}$, such that noise is distributed in $U(\sigma_{\text{coef}} \cdot \sigma_{\text{base}})$. And the baseline noise $\sigma_{\text{base}}$ is set to:

$$\sigma_{\text{base}} = \begin{bmatrix} 0.01 & \frac{\pi}{180} & 0.03 & \frac{\pi}{180} & 0.01 & 0.03 & 0.03 & \frac{\pi}{180} & 0.01 & \frac{\pi}{180} \end{bmatrix}^{\top}.$$

The reward function is defined as a constant minus the penalties related to tracking error, vehicle instability, and collision violation:

$$r = 1 - 0.4(\delta y)^2 - 0.1(\delta \phi)^2 - 1.3|\delta v| - 0.01\omega^2 - 0.01\dot{v}^2 - 0.01\dot{\omega}^2 - 2 \cdot \mathbb{I}(\rho < 0.94),$$

where $\rho$ represents the distance between the centers of the two vehicles, calculated as $\rho = \sqrt{\Delta x^2 + \Delta y^2}$. The reference speed is set to 0.3m/s, meaning $\delta v = v - 0.3$.

The Distributional Soft Actor-critic (DSAC) (Duan et al., 2025), a model-free RL algorithm, is used to train the vehicle robot. The hyperparameters for DSAC are listed in Table 16. The tests in all scenarios are accomplished by the same networks.

All results are shown in Figure 31~49. Table 17 lists the figure index for each scenario.

Table 16: **Hyperparameters for DSAC.**

| Hyperparameter | Value |
|---|---|
| Replay buffer capacity | 1000000 |
| Buffer warm-up size | 10000 |
| Batch size | 256 |
| Discount $\gamma$ | 0.99 |
| Target network soft-update rate $\tau$ | 0.005 |
| Policy delay times | 2 |
| Temperature parameter $\alpha$ | 0.2 |
| Hidden layers in critic network | [256, 256] |
| Activations in critic network | ReLU |
| Hidden layers in subnetwork $f$ | [256, 256] |
| Activations in subnetwork $f$ | ReLU |
| Optimizer | Adam |
| Critic learning rate | $1 \cdot 10^{-4}$ |
| Actor learning rate | $1 \cdot 10^{-4}$ |
| Coefficient $\lambda_k$ | 0.1 |
| Coefficient $\lambda_h$ | 0.04 |
| Length of historical obsv. $N$ | 20 |

Table 17: **Figure indices for the results of mini-vehicle driving environment.**

| Scenario and network | | Noise amplitude | | Snapshots |
|---|---|---|---|---|
| | | 0 | 10 | |
| Scenario 1 | MLP | Figure 31 | Figure 33 | Figure 35 |
| | LipsNet++ | Figure 32 | Figure 34 | |
| Scenario 2 | MLP | Figure 36 | Figure 38 | Figure 40 |
| | LipsNet++ | Figure 37 | Figure 39 | |
| Scenario 3 | MLP | Figure 41 | Figure 43 | Figure 45 |
| | LipsNet++ | Figure 42 | Figure 44 | |
| Scenario 4 | MLP | Figure 46 | Figure 48 | URL [2] |
| | LipsNet++ | Figure 47 | Figure 49 | |

Table 18: **Performance summary in mini-vehicle driving environment.**

| Task setting | | Scenario 1 | | Scenario 2 | | Scenario 3 | |
|---|---|---|---|---|---|---|---|
| Policy network | Noise amplitude | TAR | AFR | TAR | AFR | TAR | AFR |
| LipsNet++ | 0 | 234.7 | **0.02** | 252.6 | **0.04** | 287.5 | **0.03** |
| | 1 | 235.2 | **0.02** | **252.0** | **0.04** | 288.5 | **0.03** |
| | 5 | **232.8** | 0.08 | **254.1** | 0.08 | **289.6** | 0.08 |
| | 10 | **233.6** | 0.14 | **249.6** | 0.16 | **290.3** | 0.14 |
| | 20 | **224.5** | 0.27 | **252.7** | 0.28 | **281.3** | 0.23 |
| MLP | 0 | **238.4** | 0.04 | **254.6** | 0.17 | **293.5** | 0.15 |
| | 1 | **237.8** | 0.58 | 250.4 | 0.58 | **293.0** | 0.55 |
| | 5 | 232.7 | 1.68 | 250.0 | 1.62 | 289.6 | 1.58 |
| | 10 | 225.0 | 2.03 | 247.2 | 2.24 | 283.3 | 2.17 |
| | 20 | 209.8 | 2.53 | 238.9 | 2.65 | 267.9 | 2.65 |

---

[2] https://xjsong99.github.io/LipsNet_v2

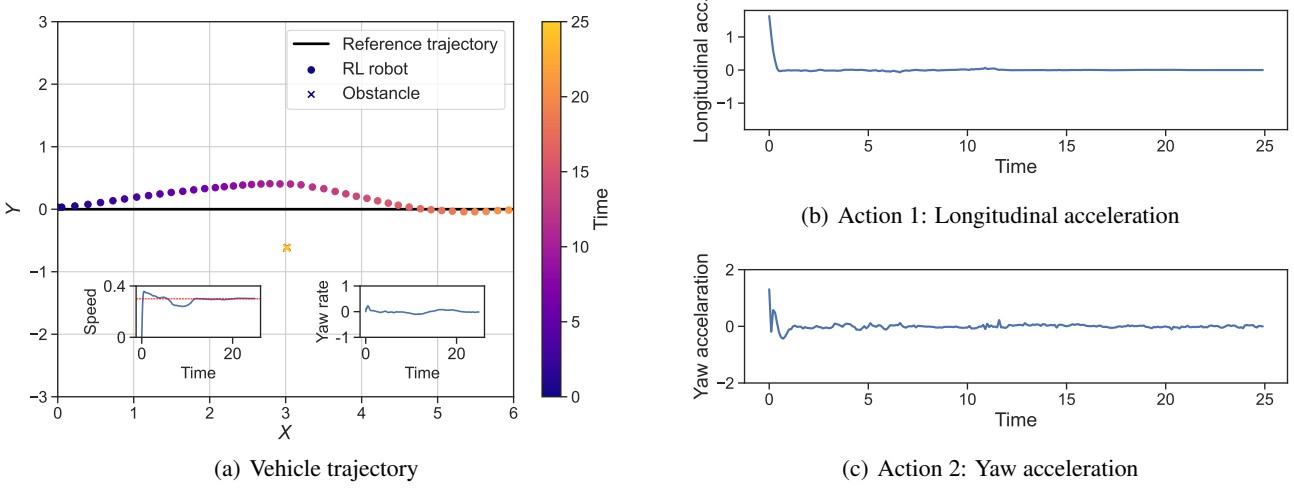

(a) Vehicle trajectory

(b) Action 1: Longitudinal acceleration

(c) Action 2: Yaw acceleration

Figure 31: **MLP performance in scenario 1.** The noise amplitude is 0.

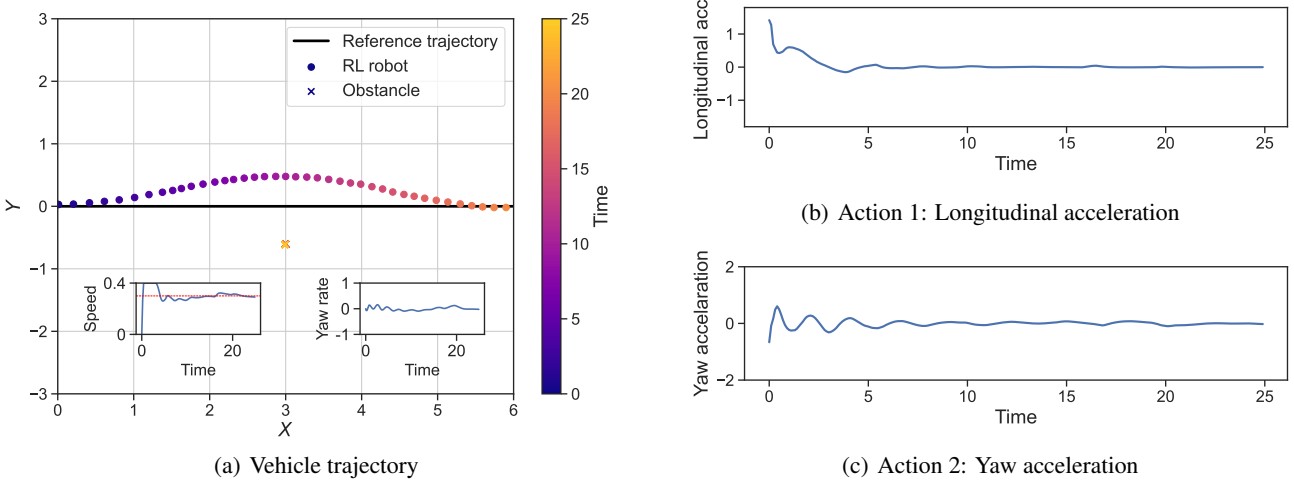

(a) Vehicle trajectory

(b) Action 1: Longitudinal acceleration

(c) Action 2: Yaw acceleration

Figure 32: **LipsNet++ performance in scenario 1.** The noise amplitude is 0.

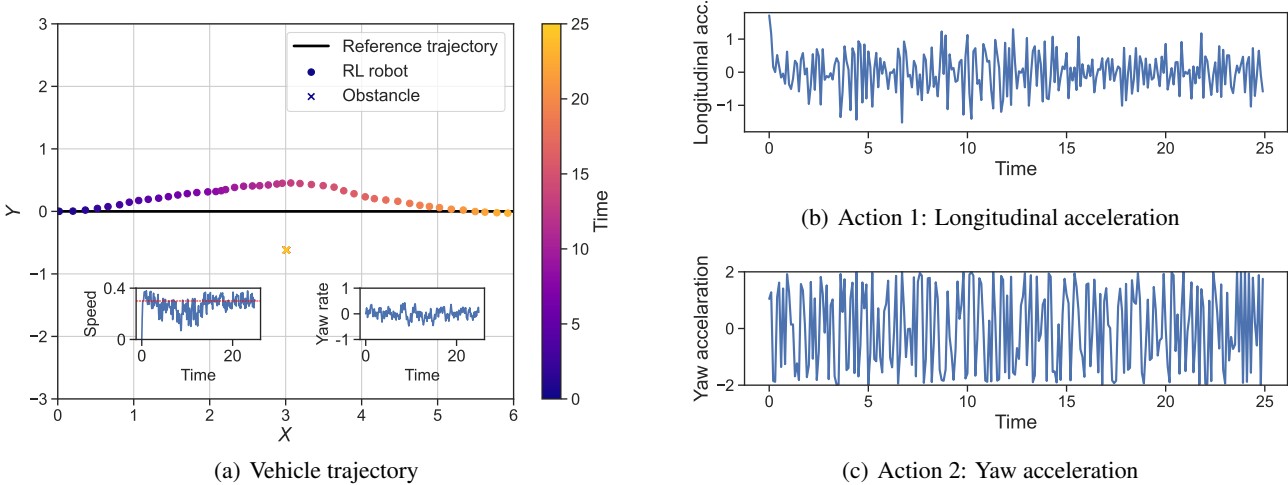

(a) Vehicle trajectory

(b) Action 1: Longitudinal acceleration

(c) Action 2: Yaw acceleration

Figure 33: **MLP performance in scenario 1.** The noise amplitude is 10.

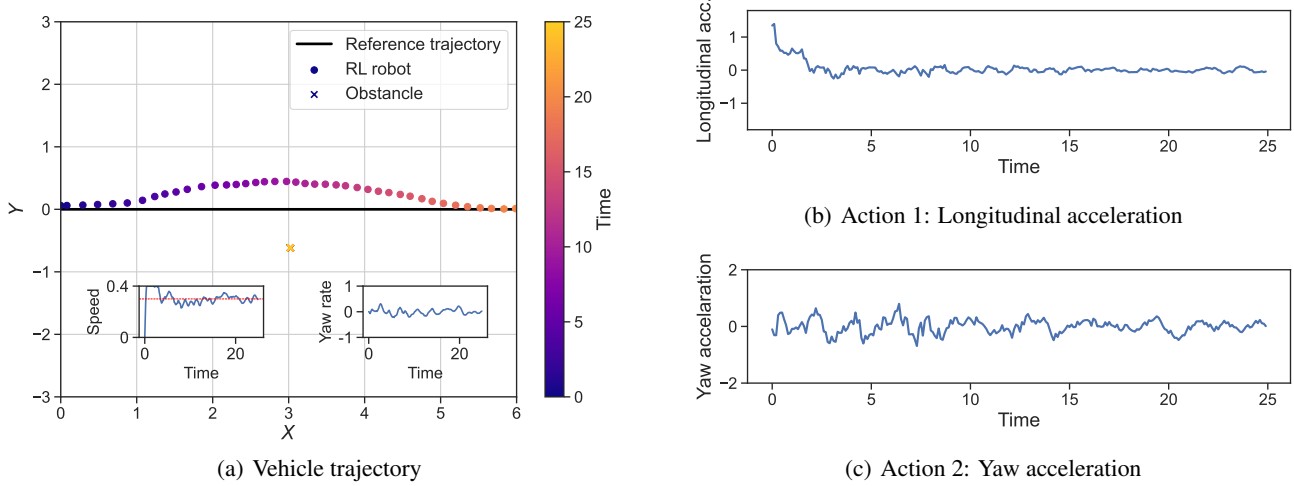

(a) Vehicle trajectory

(b) Action 1: Longitudinal acceleration

(c) Action 2: Yaw acceleration

Figure 34: **LipsNet++ performance in scenario 1.** The noise amplitude is 10.

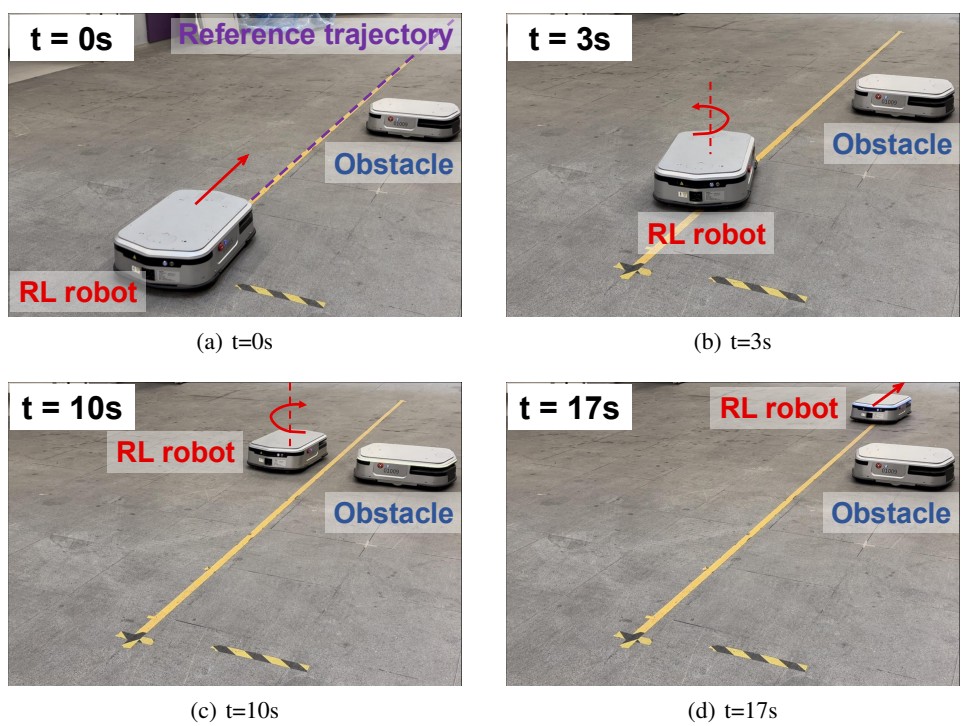

(a) t=0s

(b) t=3s

(c) t=10s

(d) t=17s

Figure 35: **Snapshots of scenario 1.**

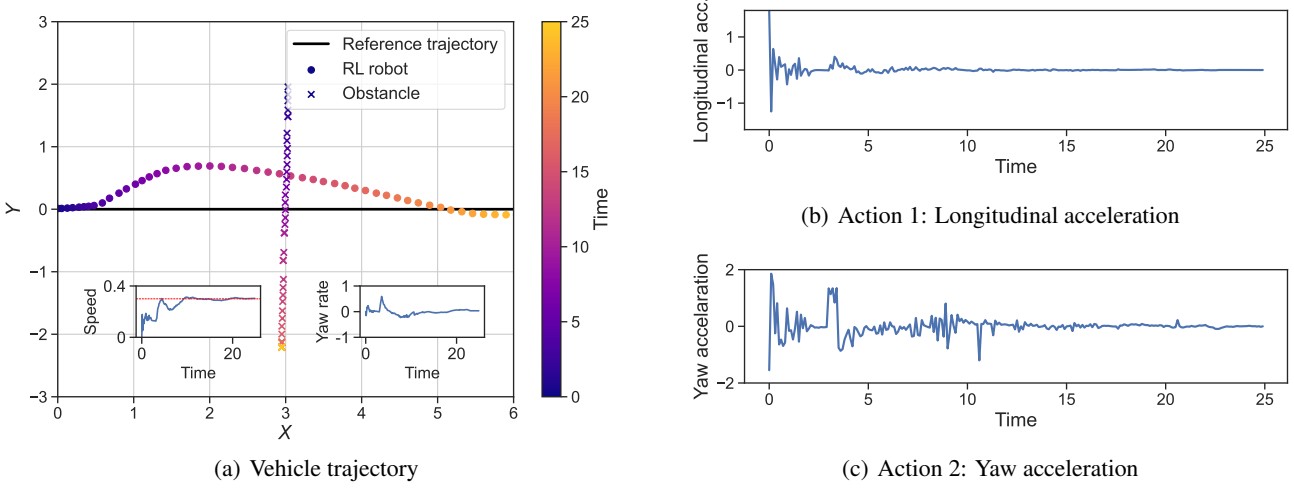

Figure 36: **MLP performance in scenario 2.** The noise amplitude is 0.

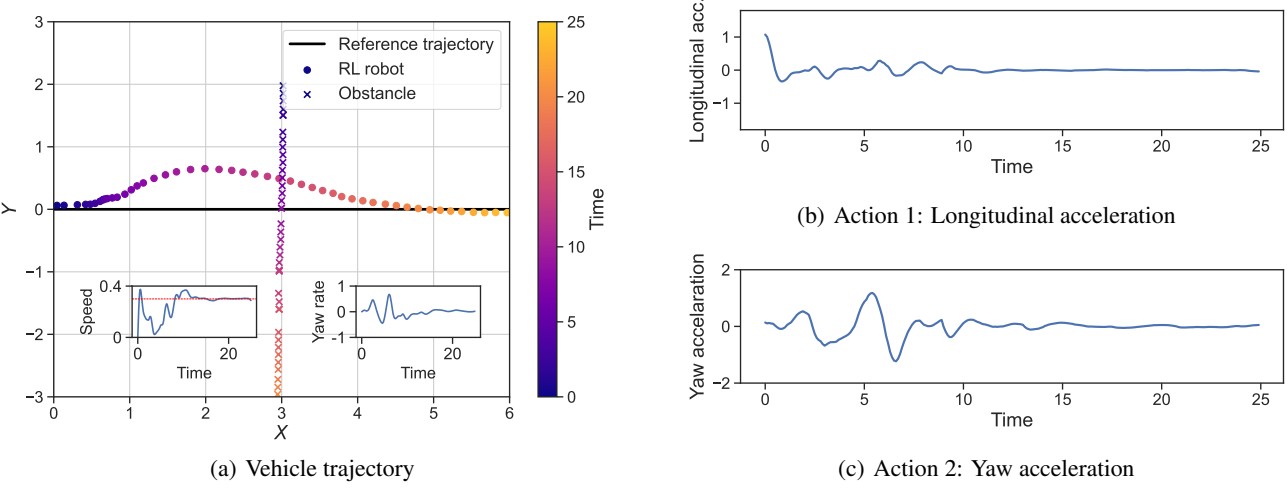

Figure 37: **LipsNet++ performance in scenario 2.** The noise amplitude is 0.

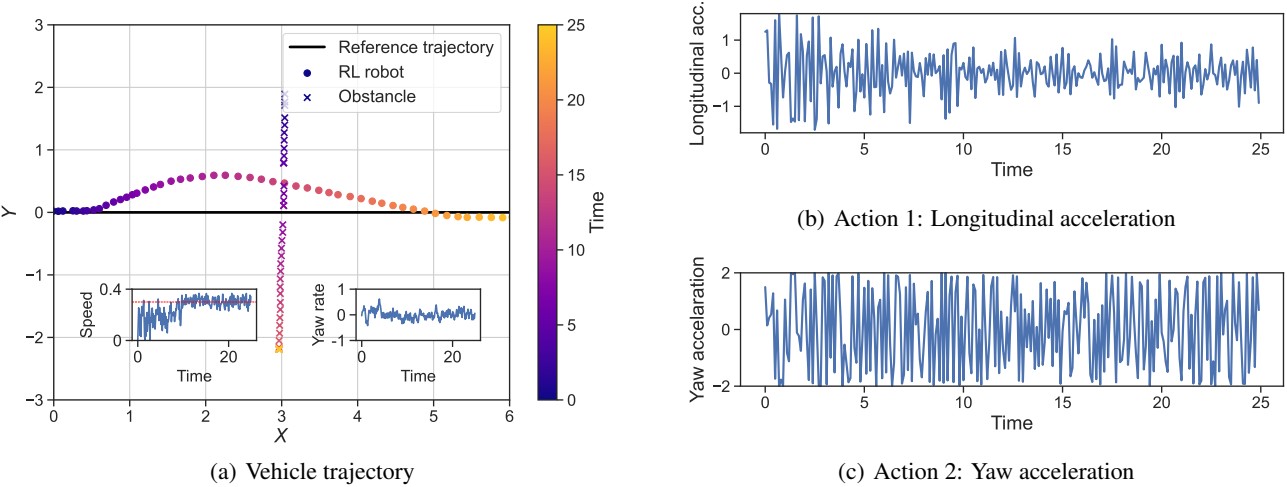

Figure 38: **MLP performance in scenario 2.** The noise amplitude is 10.

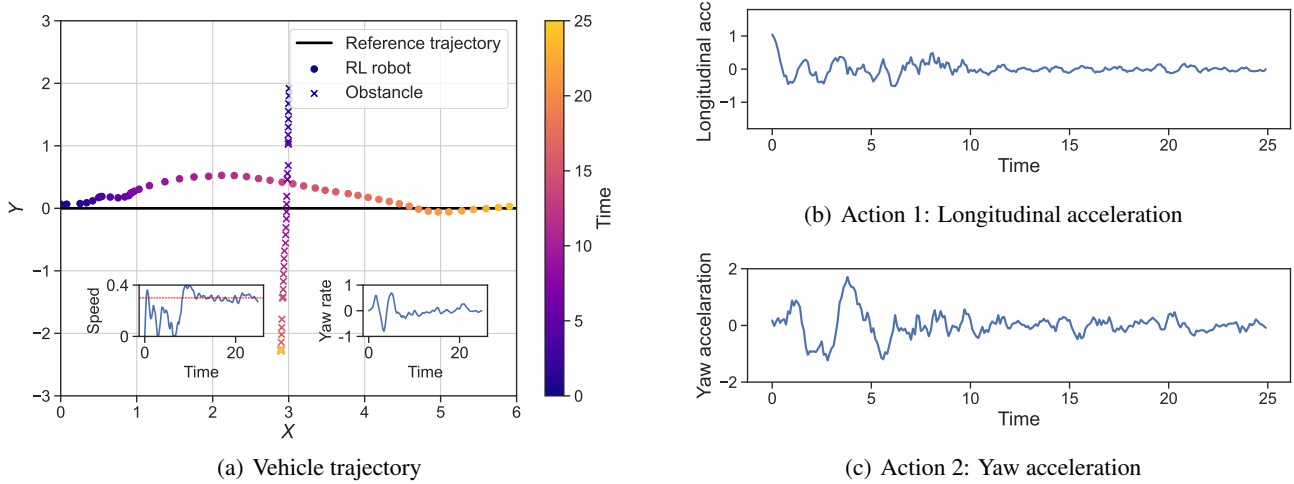

(a) Vehicle trajectory

(b) Action 1: Longitudinal acceleration

(c) Action 2: Yaw acceleration

Figure 39: **LipsNet++ performance in scenario 2.** The noise amplitude is 10.

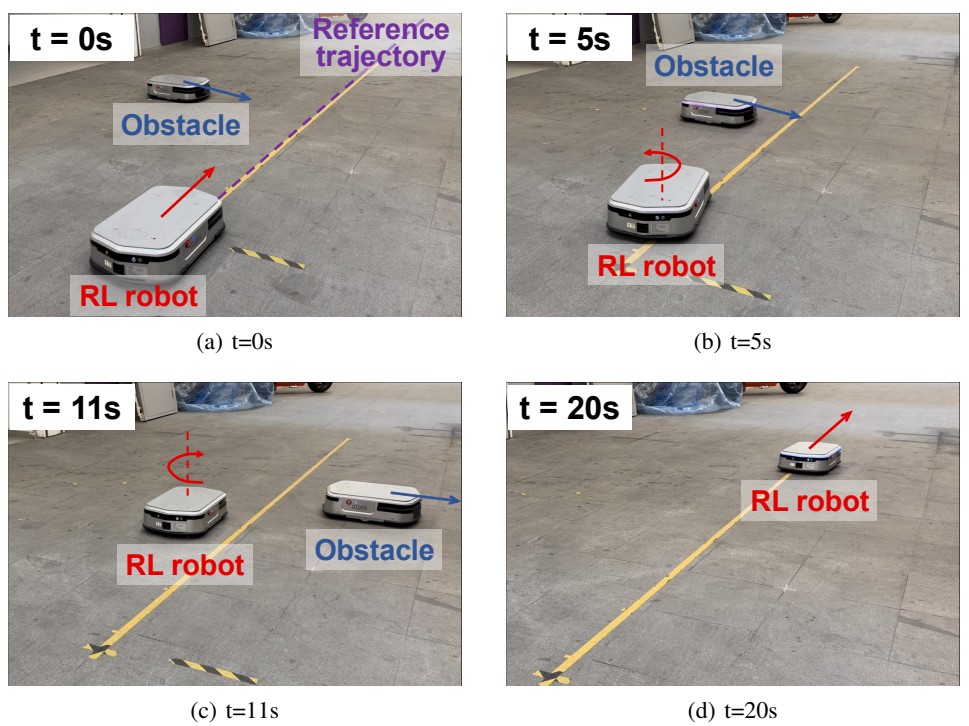

(a) t=0s

(b) t=5s

(c) t=11s

(d) t=20s

Figure 40: **Snapshots of scenario 2.**

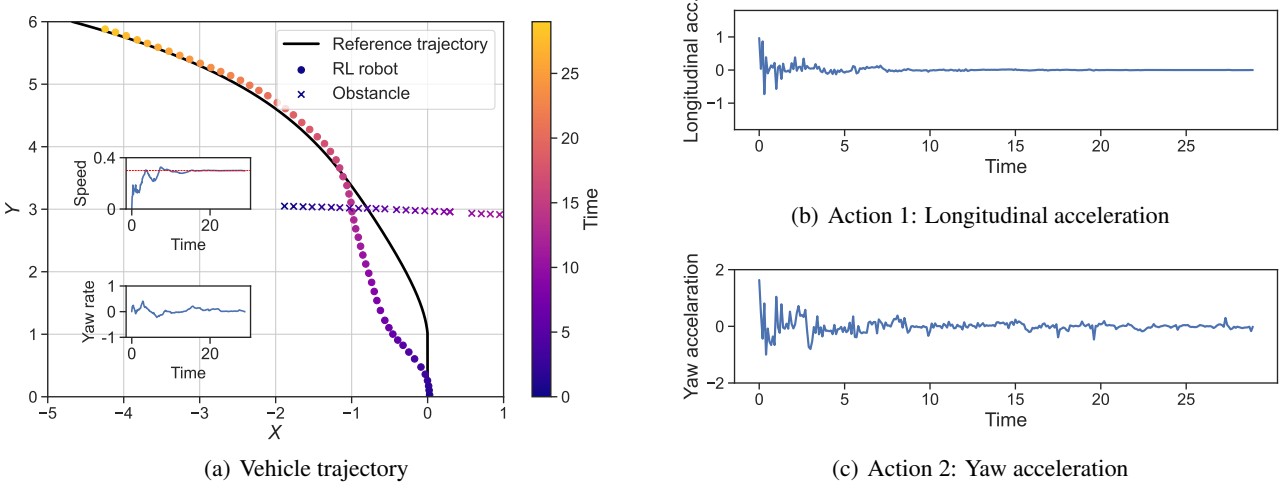

Figure 41: **MLP performance in scenario 3.** The noise amplitude is 0.

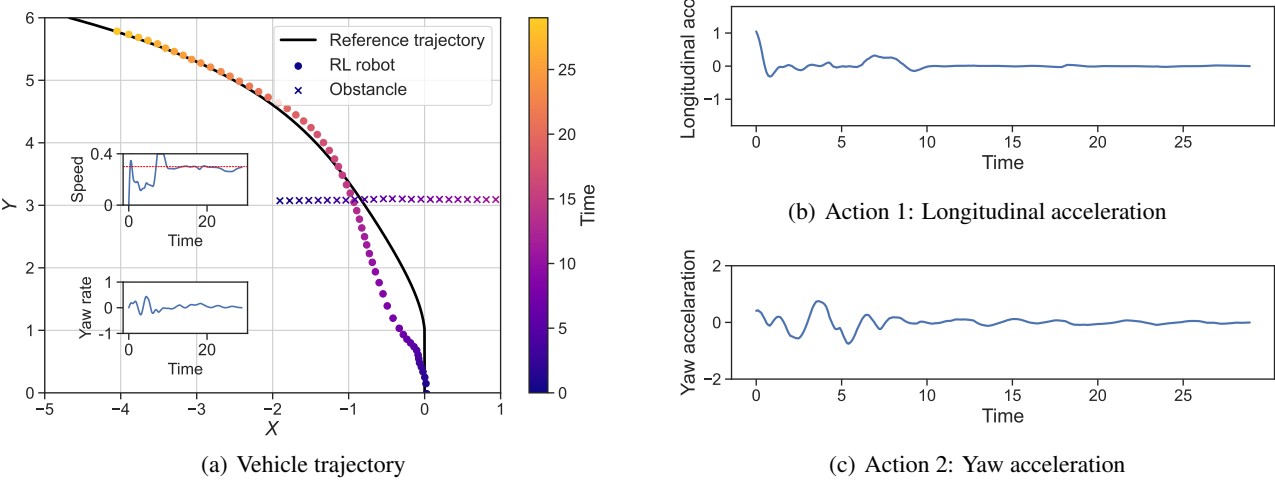

Figure 42: **LipsNet++ performance in scenario 3.** The noise amplitude is 0.

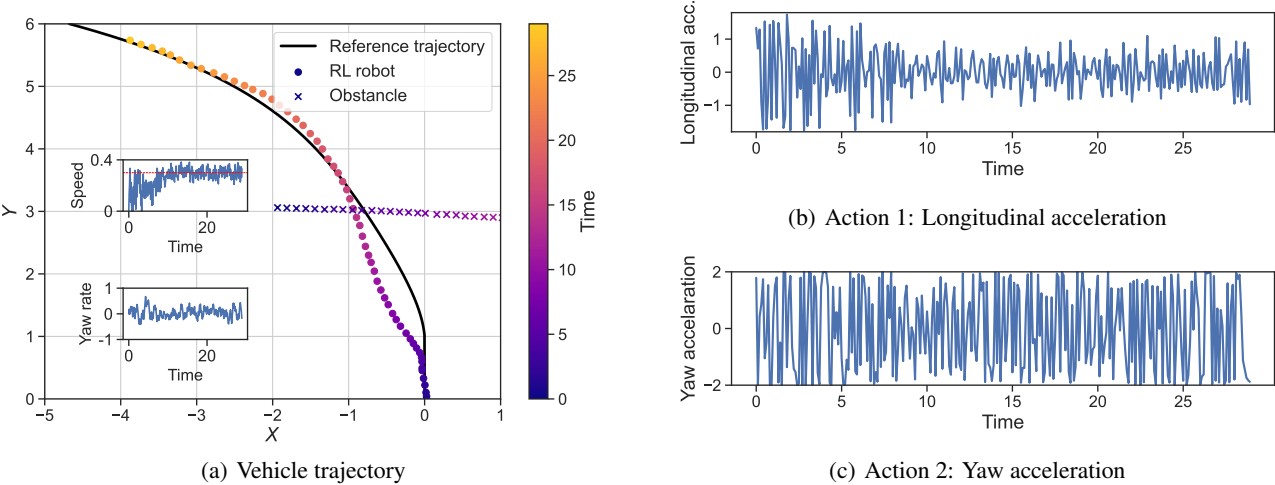

Figure 43: **MLP performance in scenario 3.** The noise amplitude is 10.

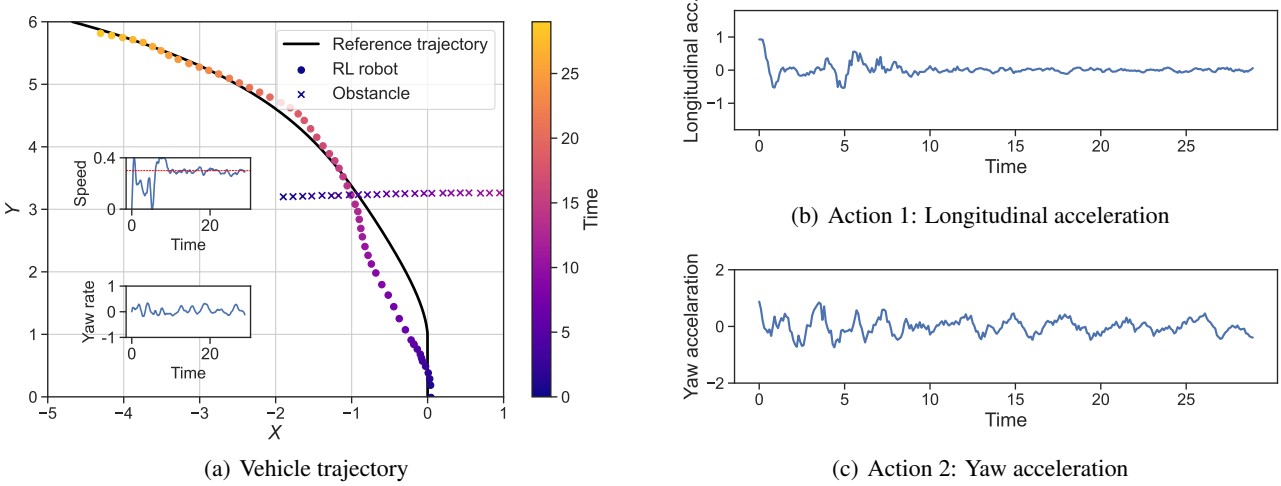

(a) Vehicle trajectory

(b) Action 1: Longitudinal acceleration

(c) Action 2: Yaw acceleration

Figure 44: **LipsNet++ performance in scenario 3.** The noise amplitude is 10.

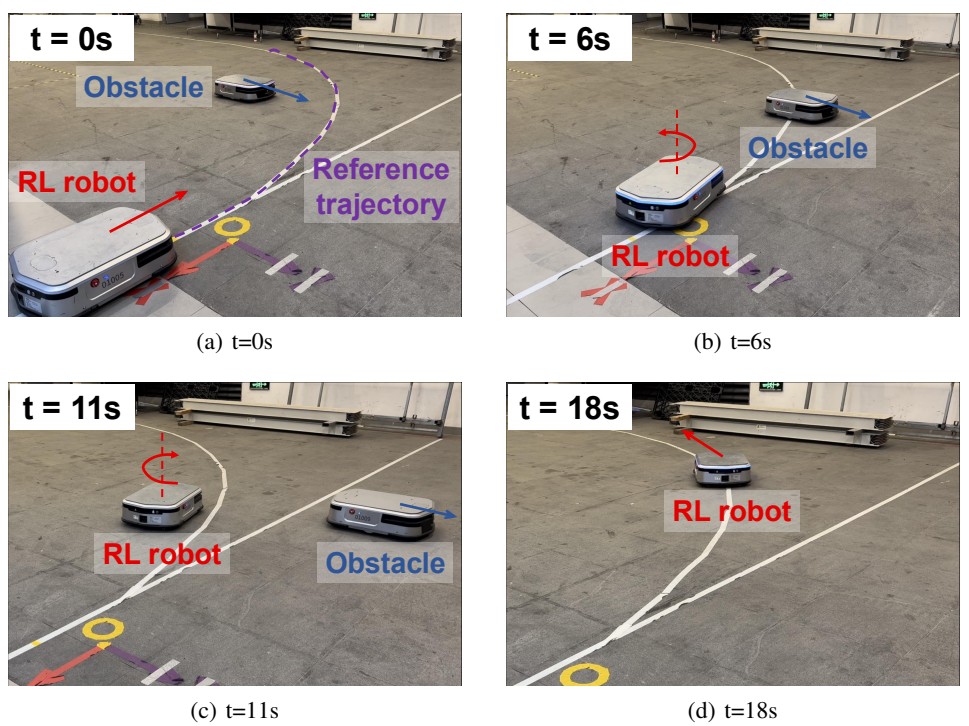

(a) t=0s

(b) t=6s

(c) t=11s

(d) t=18s

Figure 45: **Snapshots of scenario 3.**

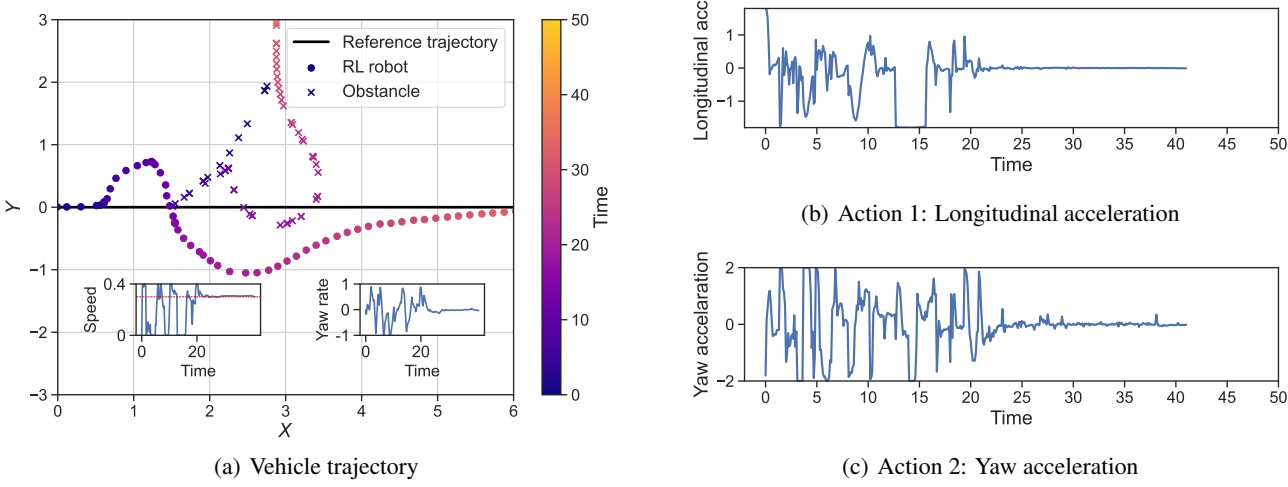

Figure 46: **MLP performance in scenario 4.** The noise amplitude is 0.

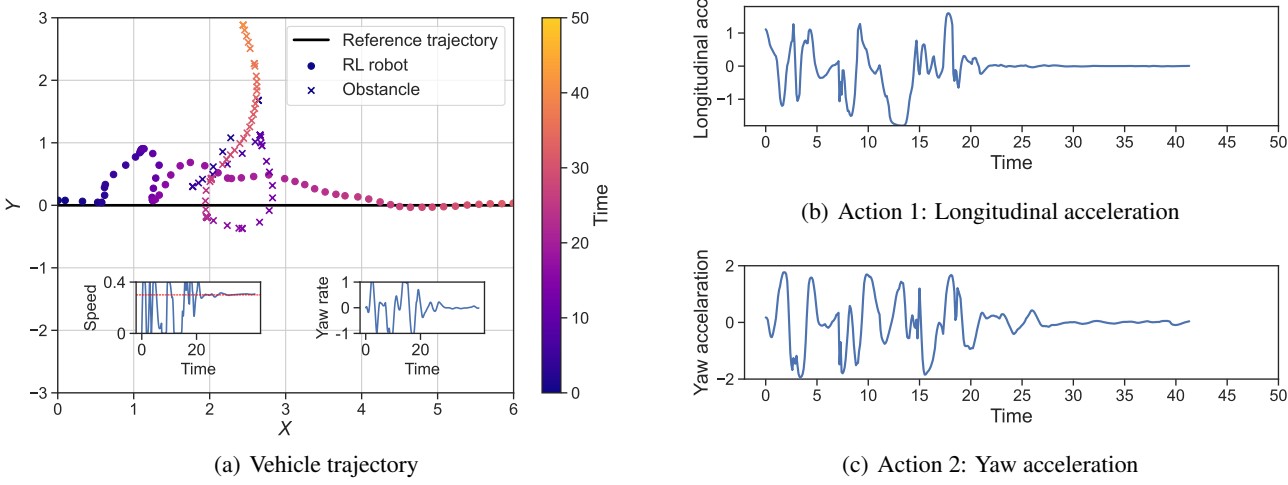

Figure 47: **LipsNet++ performance in scenario 4.** The noise amplitude is 0.

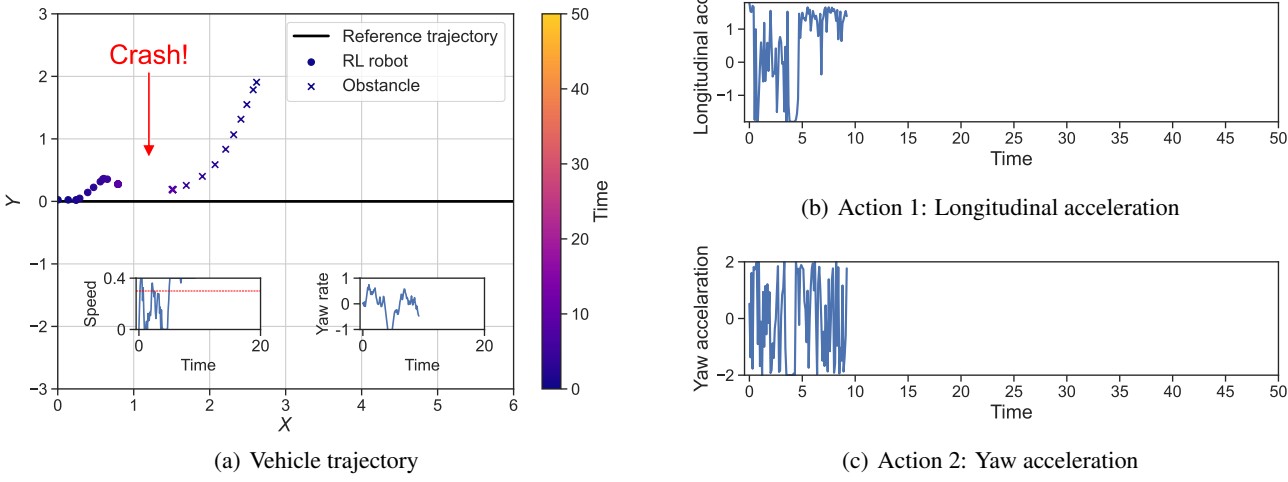

Figure 48: **MLP performance in scenario 4.** The noise amplitude is 10.

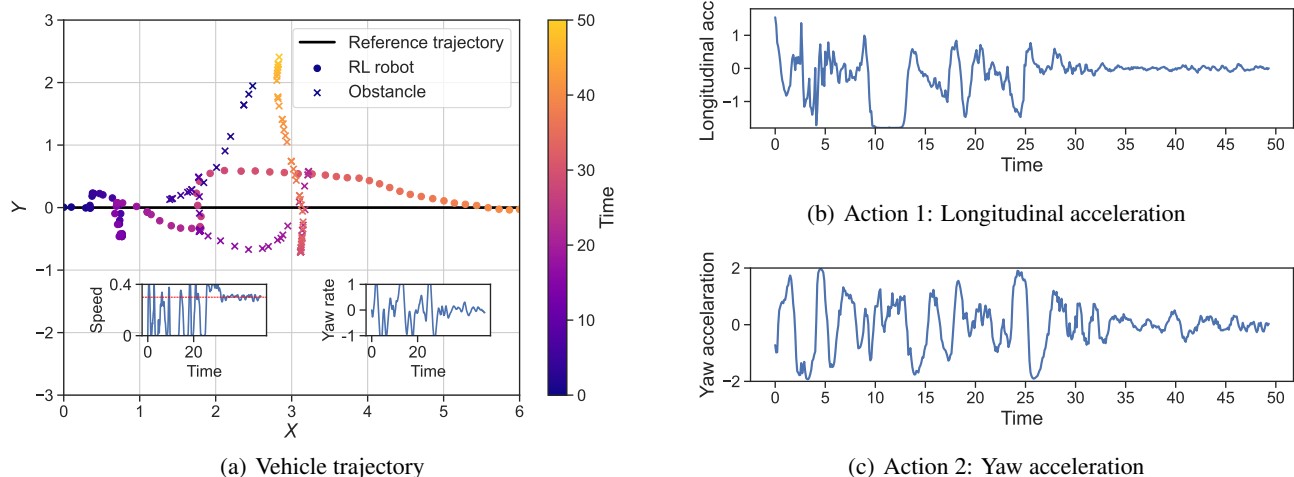

(a) Vehicle trajectory

(b) Action 1: Longitudinal acceleration

(c) Action 2: Yaw acceleration

Figure 49: **LipsNet++ performance in scenario 4.** The noise amplitude is 10.

The results of TAR and AFR for scenario 1-3 are listed in Table 18. The result for scenario 4 is not listed because the obstacle vehicle is manipulated by human, which means each trial has great randomness. The data in Table 18 is visualized in Figure 50. As shown in Figure 50(a)(c)(e), when noise increases, LipsNet++ maintains the highest TAR and its TAR declines much slower than MLP's. As shown in Figure 50(b)(d)(f), when noise increases, LipsNet++ maintains the lowest AFR and its AFR grows much slower than MLP's. These results imply LipsNet++ has excellent action smoothness and noise robustness.

## N. Limitation and Future Work

LipsNet++ achieves smoother and more robust control with a slight increase in training time, as illustrated in Appendix H. The increase in training wall time caused by this limitation becomes acceptable due to the significant reduction in action fluctuation rate.

In the future, the backward time of LipsNet++ can be optimized to overcome the above limitation. It can be accelerated by using multiple forward propagation and zero-order gradient estimation to compute the Jacobian matrix.

Furthermore, an attention mechanism for the filter matrix $H$ can be introduced in future works. In this way, $H$ can vary according to different observation inputs. Additionally, employing LipsNet++ on more real-world tasks, such as highway vehicles and embodied AI, is also a promising direction.

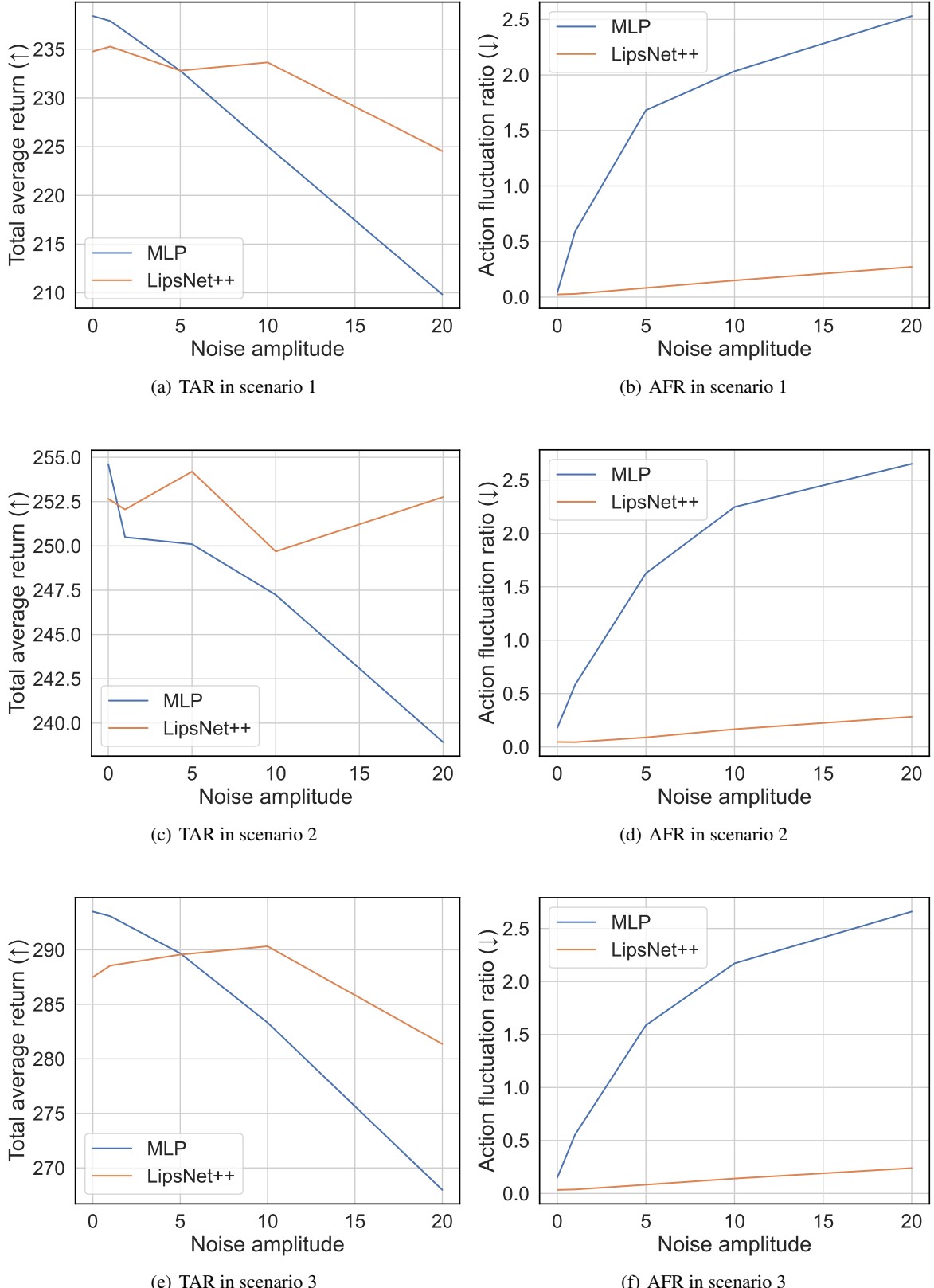

Figure 50: **Performance trend with increasing noise in mini-vehicle driving environment.**

