# OpenReview forum: "LipsNet++: Unifying Filter and Controller into a Policy Network"
_ICML.cc/2025/Conference — ICML 2025 spotlightposter_

### Official Review · Reviewer_qhMJ · 2025-03-11

**Overall Recommendation:** 3

**Summary:**

Deep reinforcement learning suffers from the action fluctuation problem. This problem has been studied in many different forms in the past. Initial works constrained the Lipschitz constant of various aspects of the policy and various value functions to achieve desired smoothness. This paper proposes two methods for this problem. First, they propose smoothing observation noise by frequency domain filters with the filtering matrix learnt via gradient descent. Second, they propose to penalize the norm of the Lipschitz constant of the neural networks. Experimental results are shown on standard reinforcement learning benchmarks.

**Claims And Evidence:**

The paper proposes two methods to smooth the actions of RL policies. They are a little ad-hoc but there is some experimental evidence to suggest that the methods are effective. For the first improvement on constraining the Lipschitz constant of the neural network, the authors claim certain training time improvements from the simplification. Given that the proposed regularization is network-wise and not layer-wise, it is less susceptible to training difficulty. Further, this simplification is claimed to affect performance to a lesser extent. The other method of using a learned frequency-domain filter is new and is claimed to give additional improvements. An additional claim is that they are less sensitive to hyper-parameters than the benchmarks. These improvements are illustrated on an extensive set of experimental benchmarks.

**Essential References Not Discussed:**

RL theory proposes some filtering methods such as below. Some theoretical filters could be discussed.

Hazan, Elad, Karan Singh, and Cyril Zhang. "Learning linear dynamical systems via spectral filtering." Advances in Neural Information Processing Systems 30 (2017).

**Experimental Designs Or Analyses:**

The experimental design is shown on the double integrator, deepmind control suite and mini-vehicle driving. The mini-vehicle driving is illustrated on physical vehicles. The experimental design is sound.

**Methods And Evaluation Criteria:**

The methods are justifiable and the evaluation criteria of the action fluctuation ratio is standard for these benchmarks. However, given the control applications of this paper, the impact of filtering and normalizing the Lipschitz constant on the stability and safety of the controllers is worrying. This suggests incorporating evaluation criteria relevant to these aspects to gain deeper understanding of the problem.

**Other Comments Or Suggestions:**

For control problems, observation filters could create a lag which can affect the system stability and safety. It would be interesting to look at these filters from a perspective of safety when the system is operating at the edge of its capability. Will smoothing the actions cause catastrophic failures when the only option is to use a big acceleration and stop to a halt? An adaptive smoothing scheme that remains cognizant of invariant constraints is much important for this line of work to expand in scope.

Section 3.2 is very interesting to me.

Line 160: The mapping function has significant output differences even if the inputs are closely adjacent

Line 162: typo with “does” not

Line 206: "However, LipsNet is not applicable in high-real-time tasks due to the Jacobian matrix calculation during forward inference," -unclear statement

Page 11, Line 557: It is unclear what is the “real Lipschitz constant”.

Theorem A.2. There are some mistakes in line 626 which should be an upper bound.

The appendix has multiple typos for “acceleration” and “obstacle”.

**Other Strengths And Weaknesses:**

Strengths

The effectiveness of the method despite its simplicity is the strength of the paper. The experimental benchmarks and figures are strong. The results are not compelling on all benchmarks. Further, experiments with adversarial noise could be interesting to evaluate further.  The result in Figure 12 is strong.

Weaknesses

The structure and presentation is very similar to Lipsnet. The Lipschitz constant optimization of Lipsnet has more rigor.

Only one of the two proposed methods fit the criteria of being high on both novelty and impact. These weaknesses are reflected in my score for this paper even though the experimental results have strength.

**Questions For Authors:**

In terms of the claims of robustness, do we know if the proposed method is better or worse on adversarial noise distributions?

Is the noise distribution same in training and test?

What aspects of the training was changed during the test?

**Relation To Broader Scientific Literature:**

Action smoothness is a very important problem to be tackled for control and robotics applications. The frequency domain filtering method is interesting and deserves mention. The other method is similar to previously known literature and is not a standout approach.

**Theoretical Claims:**

The theoretical section in the appendix is not very new and only serves as an addendum to the reader. In theorem A.2, it should be the case that $K(x)\leq max_{x^{\prime}} \Vert \nabla f(x) \Vert$. The approximation in the theorem statement is concerning given that there is theoretical research on the lipschitz constant of a neural net.

---

> ### Author Rebuttal · Authors · 2025-03-31
>
> Thank you for your insightful feedback!
> We are encouraged by the positive aspects of your assessment, i.e., ***method effectiveness, sound experimental design, and strong benchmarks and figures***. Your recognition of the frequency-domain filtering, Section 3.2, and Fig. 12 is especially appreciated.
>
> We now address your comments in detail:
>
> ---
> #### **> Improved Theoretical Claims**
> In Line 626, **the equality $K=\max_{x'}\left\Vert\nabla f(x')\right\Vert$ follows from Equation (8) in Lemma A.1**, rather than a direct application of the mean value theorem. So, the inequality $\leq$ does not apply here. To ensure a more rigorous presentation, we have explicitly included the error term into the theorem in the main text and discussed the limiting case $\rho\to0$. Please refer to our response to reviewer QXnu for the revised theorem.
>
> #### **> Improved Discussion with Additional References**
> Thank you for pointing out some theoretical filters, such as [1-3]. These works exhibit strong theoretical foundations. In contrast, our approach is more experimental—**we don't make any assumptions about system linearity, convexity, or noise distribution.** Instead, we introduce a learnable filtering matrix that **adaptively learns from data which frequencies are important and which are noise**. We have added a detailed discussion in introduction section and don't elaborate here due to space constraints.
>
> #### **> Weakness Concerns**
> In fact, LipsNet++ differs significantly from LipsNet in structure. **First**, LipsNet only addresses excessive Lipschitz constant, overlooking the dual causes of action fluctuation. In contrast, our approach introduces the Fourier Filter Layer and Lipschitz Controller Layer, which not only control the Lipschitz constant but also directly address observation noise. **Second**, while LipsNet has a rigorous Lipschitz constraint, it relies on a strong assumption—only for MLP with piece-wise linear activations (ReLU). Our Jacobian regularization, however, generalizes to any network architectures. **Third**, LipsNet’s Lipschitz constraint method is too slow for real-time inference, whereas LipsNet++ achieves a 4× faster inference speed, as shown in Appendix H. In summary, the two techniques in our work are both highly novel and impactful, with a structure entirely distinct from LipsNet.
>
> #### **> Comments & Suggestions**
> The lag when operating at the edge of capability is an interesting topic. Notably, LipsNet++ prevents catastrophic failures when the only option is to use a big acceleration and stop to a halt. The DMControl-Cheetah environment exemplifies this scenario, requiring actions to switch rapidly between -1 and 1 for high-speed movement. The [[Link - Figure R1]](https://github.com/ICML-anonymous-2025/LipsNet_v2/blob/main/rebuttal/Figure_R1.png) visualizes the first two action dimensions (rapid switching occurs across all dimensions). As shown in the figures, LipsNet++ preserves the necessary rapid reaction behavior to maintain good control performance (see TAR in Fig. 9) while reducing action fluctuation as much as possible (see AFR in Fig. 9). LipsNet++ **can adapt action smoothness dynamically** by learning local Lipschitz constants that vary with different obs input.
>
> #### **> Fixed Typos & Unclear Words**
> Line 160: Typo fixed (*even* $\to$ *even if*)
>
> Line 262: Typo fixed (*dose* $\to$ *does*)
>
> Line 206: We have clarified this statement by explicitly noting that LipsNet’s Jacobian matrix calculation during inference introduces significant computational overhead. The inference time and their comparisons are already detailed in Appendix H.
>
> Line 557: Fixed by removing *'real'*
>
> Appendix typos: Fixed all typos of *'acceleration'* and *'obstacle'*
>
> #### **> Questions**
> Q1: We have not tested with adversarial noise, which is a promising direction for future work.
>
> Q2 & Q3: Training is conducted without noise. Our method doesn't require the presence or absence of noise during training. During testing, different levels of noise are added to evaluate performance and action fluctuation under varying noise intensities. Apart from this, the training and test setups remain identical. The ability to achieve filtering even with noise-free training is attributed to Equation (4): the term $\lambda_h\left\Vert H\right\Vert_F$ drives the filter coefficients of frequencies unrelated to control performance to 0, as shown in Fig. 11, 22, and 28. This mechanism is one of our core ideas.
>
> ---
> Thank you again for your time and valuable feedback! Please let us know if you have any further questions.
>
> #### ***References***
> *[1] H. Elad, K. Singh, and C. Zhang. Learning linear dynamical systems via spectral filtering. NeurIPS 2017.*
>
> *[2] H. Elad, H. Lee, K. Singh, C. Zhang, and Y. Zhang. Spectral filtering for general linear dynamical systems. NeurIPS 2018.*
>
> *[3] A. Sanjeev, E. Hazan, H. Lee, K. Singh, C. Zhang, and Y. Zhang. Towards provable control for unknown linear dynamical systems. ICLR Workshop 2018.*

---

> > ### Comment · Reviewer_qhMJ · 2025-04-03
> >
> > Thank you for the detailed response to my comments.
> >
> > I read the improved theorem statement and the changes are important for theorem to be rigorous. It is better to avoid using an approximate operator without further justification.
> >
> > In terms of training conducted with and without noise, I am still curious about this aspect. Adding noise or disturbance can excite additional modes of the underlying dynamics that were previously not excited. This can have a significant impact on training and the learned policy. The frequencies related to the control performance depend on the noise distribution if the system is not fully observable. In this paper, we assume perfect knowledge of the state which is not known usually.

---

> > > ### Author Response · Authors · 2025-04-06
> > >
> > > Thank you very much for your prompt response!
> > >
> > > Avoiding using an approximate operator has indeed made our theorem more rigorous, and we have revised the paper accordingly.
> > >
> > > We appreciate your insightful question on training noise—this is a valuable point.
> > > This reply box provides us with more characters to further elaborate on this question.
> > > **In short, LipsNet++ can be trained under any noise level, including noise-free settings.**
> > > We will substantiate this claim through both theoretical analysis and empirical evidence:
> > >
> > > #### **> Theoretical Analysis**
> > > As shown in Equation (3), the filtered frequency amplitude is determined by the magnitude of the complex elements in the Hadamard product $X\odot H$, where $X$ is the frequency feature matrix of the input observations and $H$ is the learnable filter matrix.
> > > If the magnitude of an element in $H$ approaches zero, it means that the corresponding frequency will be suppressed.
> > >
> > > LipsNet++ can **automatically identify valuable frequencies and noise frequencies**, thanks to the actor loss in Equation (4):
> > > $$\hspace{6cm}\mathcal{L}'=\mathcal{L}+\lambda_h\Vert H\Vert_F,$$
> > > where $\mathcal{L}$ is the origin actor loss, $\Vert H\Vert_F$ is the Frobenius norm,
> > > and $\lambda_h$ is a coefficient.
> > >
> > > (1) The first term $\mathcal{L}$ aims to improve the policy control performance (total average return) as much as possible.
> > >
> > > (2) The second term $\Vert H\Vert_F$ aims to decrease the magnitude of elements in $H$ as much as possible.
> > >
> > > Due to the **mutual influence** of the above two terms, **the filter coefficients (element magnitudes) in $H$ for frequencies that do not affect performance will automatically decrease to zero, while the filter coefficients for important frequencies that do affect performance will remain at higher values.**
> > > In this way, the automatic extraction of important frequencies and the suppression of non-important frequencies are achieved.
> > > This analysis does not involve the presence or absence of training noise, so LipsNet++ can be trained under any noise level.
> > >
> > > You mentioned, *"Adding noise or disturbance can excite additional modes,"* and we fully agree.
> > > For example, noise is often added in RL to enhance exploration.
> > > LipsNet++ can also be trained with noise (i.e., when the system is not fully observable), but training with noise does not affect its control performance or action smoothing ability.
> > > In addition to the reason mentioned above, this is because we use **2D FFT** filtering in the Fourier Filter Layer, as shown in Equation (2) and (5), rather than applying 1D FFT filtering directly to each observed frequency.
> > > If the noise distribution is very coincidental, 1D FFT might fail to identify valuable/noise frequencies.
> > > However, 2D FFT works fine because the **interdependencies between observation dimensions** during filtering allow for **more precise identification of whether an observation change is due to noise or a system state variation**. For example, when speed and position exhibit **mutually consistent changes**, 2D FFT can infer that the vehicle is accelerating rather than experiencing noises of the position sensor and speed sensor; otherwise, the observed changes are likely caused by noise from one of the sensors.
> > >
> > >
> > > #### **> Empirical Evidences**
> > > To validate the above conclusion, we added **comparative training** under **noise-free, uniform noise, and normal noise** conditions in the mini-vehicle driving environment, and **visualized** the resulting filter matrices $H$.
> > >
> > > The results are shown in [[Link - Figure R2]](https://github.com/ICML-anonymous-2025/LipsNet_v2/blob/main/rebuttal/Figure_R2.png).
> > >
> > > These experimental results show that, regardless of noise presence or distribution, **the mode of the trained filter matrices remains consistent**, as only a fixed set of observation frequencies affect performance. Intuitively, the matrix $H$ can be seen as a form of *attention*: once the model learns which frequencies are worth focusing on, it naturally ignores the noisy ones; therefore, the performance remains unaffected when tested in noisy environments. This illustrates one of the **advantages of frequency-based processing**—training under any noise level (including noise-free settings) allows $H$ to identify the key frequencies.
> > >
> > > Finally, we acknowledge that under certain adversarially designed noise, the noise frequencies may overlap with the key frequencies, which calls for future works to deal with. **In general conditions, LipsNet++ can be trained under any noise level, including noise-free settings.**
> > >
> > > ---
> > > We hope our reply adequately address your concern.
> > >
> > > All the above analysis have been incorporated into the revised paper. We must emphasize that your valuable feedback has significantly improved the paper's quality.
> > >
> > > Thank you sincerely for your time and valuable feedback!

---

### Official Review · Reviewer_K8nQ · 2025-03-14

**Overall Recommendation:** 3

**Summary:**

The paper proposes a new policy network, LipsNet++, to mitigate the action fluctuation problem in real-world robotic applications. The proposed network uses a Fourier filter layer to smooth the observation input which is then fed into a MLP network with its local Lipschitz constant regularized via Jacobian regularization. On a set of simulated and real robotic tasks, the proposed policy network exhibits a lower action fluctuation level and higher robustness to observation noises. In particular, on the double integrator task, the authors compare LipsNet++ against a set of prior approaches designed for mitigating the action fluctuations and show that the proposed network excels at all metrics, including control performance and action smoothness when subject to different amount of observation noises.

## update after rebuttal
All my concerns have been addressed. With the new experiments, I am now convinced that LipsNet++ indeed outperforms previous approach on the action smoothness while not sacrificing performance.

I have increased my score from 1 to 3.

**Claims And Evidence:**

"Simulated and real-world experiments show that LipsNet++ has excellent action smoothness and noise robustness, achieving a new SOTA performance." -- I am not convinced by this claim because the only environment where the authors compare LipsNet++ with the prior baseline approaches (CAPS, L2C2, MLP-SN) is the toy double integrator environment. It is unfair to say that LipsNet++ achieves SOTA when it is only really shown to be the best on a toy environment.

**Essential References Not Discussed:**

- One of the key component of the proposed policy network is the jacobian regularization of the network. However, this exact technique has been used in many prior works (especially in adversarial robustness literature, e.g., [1, 2].) but the paper fails to attribute this technique properly
- The other key component of the proposed policy network is the observation filtering with the goal of being robust to observation noises. This is also widely studied in the literature (e.g., [3, 4]). The paper also fails to discuss how the proposed technique relates to prior techniques.

While the ideas in the paper are nice, it is hard to contextualize this paper in the literature without proper discussions with the broader literature.

[1] Hoffman, Judy, Daniel A. Roberts, and Sho Yaida. "Robust learning with jacobian regularization." arXiv preprint arXiv:1908.02729 (2019).
[2] Jakubovitz, Daniel, and Raja Giryes. "Improving dnn robustness to adversarial attacks using jacobian regularization." Proceedings of the European conference on computer vision (ECCV). 2018.
[3] Zhang, Huan, et al. "Robust deep reinforcement learning against adversarial perturbations on state observations." Advances in Neural Information Processing Systems 33 (2020): 21024-21037.
[4] Liu, Zuxin, et al. "On the robustness of safe reinforcement learning under observational perturbations." arXiv preprint arXiv:2205.14691 (2022).

**Experimental Designs Or Analyses:**

The experimental results and analyses are well written and convincing. My only concern is that the authors did not compare the proposed method against important baselines on environments other than the toy double integrator environment. This raises the important question that whether the proposed method is as effective as claimed to be (SOTA).

**Methods And Evaluation Criteria:**

The proposed method makes sense and the evaluation criteria are also reasonable.

**Other Comments Or Suggestions:**

N/A

**Other Strengths And Weaknesses:**

Strengths:
- The paper is easy to follow and the proposed method makes sense for addressing the action fluctuation problem.

Weaknesses:
- The empirical comparison of the prior methods is lacking. While the authors evaluate the proposed method on many environments, baseline results are only available on a toy double integrator environment. This alone is not sufficient to show the effectiveness of the proposed method.

**Questions For Authors:**

N/A

**Relation To Broader Scientific Literature:**

The paper is closely related to LipsNet with the same goal: mitigating action fluctuation in robotic applications. The key contributions of this paper are
- Observation filtering module with a learnable filter matrix that can help smooth out noises in the observations.
- Jacobian regularization to encourage the local Lipschitz constant to be small for the policy network

**Theoretical Claims:**

The statement for Theorem 3.2 is very vague -- What does "$||\nabla_x f|| \approx K(x)$" mean? I would suggest to use a more precise statement (e.g., define what infinitesimal neighborhood means and state that as $\rho \rightarrow 0,  ||\nabla_x f|| \rightarrow K(x)$).

---

> ### Author Rebuttal · Authors · 2025-04-01
>
> Thank you for your insightful feedback!
> We appreciate your recognition of our ***well-written and convincing*** results and analyses, ***reasonable*** evaluation criteria, ***nice*** ideas, the proposed methods that ***make sense***, and the ***easy-following*** nature of our paper. The positive aspects of your assessment encourages us.
>
> We now address your comments in detail:
>
> ---
> #### **> Improved Theoretical Claim**
> We appreciate your valuable suggestion! We **have refined** Theorem 3.2 by explicitly **incorporating the approximation error, defining the neighborhood area, and describing its asymptotic behavior** when $\rho\to 0$, as you suggested. The revised theorem is as follows:
>
> **Theorem 3.2** (Lipschitz’s Jacobian Approximation): *Let $f:\mathbb{R}^n \to \mathbb{R}^m$ be a continuously differentiable neural network. The Jacobian norm $\left\Vert\nabla_xf\right\Vert$ serves as an approximation of the local Lipschitz constant of $f$ within the neighborhood $\mathcal{B}(x, \rho)$, centered at $x$ with radius $\rho$. The approximation error is given by $\mathop{\max}_{\ \delta\in \mathcal{B}(0,\rho)} \left[ \left(\nabla_x\left\Vert \nabla_x f(x) \right\Vert\right)^\top\delta + o(\delta)\right].$ Moreover, as $\rho \to 0$, the Jacobian norm converges to the exact local Lipschitz constant, i.e. $\left\Vert\nabla_xf\right\Vert \to K(x)$.*
>
> #### **> Additional Comparisons with Baselines on DMControl**
> Beyond the comprehensive baseline comparisons in the double integrator environment, Section 4.2 evaluates LipsNet++ against three baselines (MLP, LipsNet-G, LipsNet-L) across all DMControl environments and Appendix J compares it with one baseline (MLP-SN) on Reacher environment. Extending MLP-SN to all environments is impractical due to excessive hyperparameter tuning (Line 1039).
>
> To further address your concerns, we **have added experiments with additional baselines on DMControl** environments. Now, LipsNet++ is evaluated against all baselines on DMControl, including MLP, CAPS, L2C2, MLP-SN, LipsNet-G, and LipsNet-L.
>
> TAR comparison: [[Link - Table R1]](https://github.com/ICML-anonymous-2025/LipsNet_v2/blob/main/rebuttal/Table_R1.png), AFR comparison: [[Link - Table R2]](https://github.com/ICML-anonymous-2025/LipsNet_v2/blob/main/rebuttal/Table_R2.png)
>
> As shown in the above tables, LipsNet++ **maintains SOTA** performance on the **non-toy DMControl** tasks.
>
> #### **> Improved Discussion with Additional References**
> - Jacobian regularization: Prior works [1,2] use Jacobian regularization to enhance adversarial robustness for neural networks, with [2] proposing an efficient Jacobian norm approximation method (already cited in Line 214). While common [1-4], it is rarely explored in RL. Studies [5,6] have identified high Lipschitz constant of policy network as a cause of action fluctuation. Our Theorem 3.2 shows the Jacobian norm is an approximation of Lipschitz constant, introducing Jacobian regularization as a practical way to control Lipschitz constant in RL.
>
> - Observation filtering: Methods [7-10] enhance RL robustness in algorithm level, e.g., state-adversarial DRL with SA-MDP modeling for inaccurate observations [7] and robust training framework for safe RL with rigorous theoretical analysis and empirical results [8]. In contrast, our approach improves action smoothness and robustness in network level by designing policy network (Fourier Filter Layer), embedding prior knowledge directly into its structure.
>
> We have added a detailed discussion in introduction section and don't elaborate here due to space constraints.
>
> ---
> #### **Summary**
> We hope our revisions adequately address your concerns. Please let us know if you have any further questions.
>
> Thank you for your time and valuable feedback!
> #### ***References***
> *[1] D. Jakubovitz, et al. Improving dnn robustness to adversarial attacks using jacobian regularization. ECCV 2018.*
>
> *[2] J. Hoffman, et al. Robust learning with jacobian regularization. arXiv 2019.*
>
> *[3] W. Yuxiang, et al. Orthogonal jacobian regularization for unsupervised disentanglement in image generation. ICCV 2021.*
>
> *[4] K. Co, et al. Jacobian regularization for mitigating universal adversarial perturbations. ICANN 2021.*
>
> *[5] R. Takase, et al. Stability-certified Reinforcement Learning via Spectral Normalization, arXiv 2020.*
>
> *[6] X. Song, et al. LipsNet: A Smooth and Robust Neural Network with Adaptive Lipschitz Constant for High Accuracy Optimal Control, ICML 2023.*
>
> *[7] H. Zhang, et al. Robust deep reinforcement learning against adversarial perturbations on state observations. NeurIPS 2020.*
>
> *[8] Z. Liu, et al. On the robustness of safe reinforcement learning under observational perturbations. ICLR 2023.*
>
> *[9] M. Xu, et al. Trustworthy reinforcement learning against intrinsic vulnerabilities: Robustness, safety, and generalizability. arXiv 2022.*
>
> *[10] H. Zhang, et al. Robust reinforcement learning on state observations with learned optimal adversary. ICLR 2023.*

---

> > ### Comment · Reviewer_K8nQ · 2025-04-05
> >
> > Thanks for your response and adding the literature discussion.
> >
> > > Improved Theoretical Claim
> >
> > What does $o(\delta)$ mean here?
> >
> > > Extending MLP-SN to all environments is impractical due to excessive hyperparameter tuning (Line 1039).
> >
> > It is a common practice to tune the baselines with a similar budget as your method for comparison fairness, so I don't think it is necessary to tune MLP-SN more than what you did to tune your method. As MLP-SN seems to be a pretty strong (and simple) baseline, not having experimental results on it does bring down my confidence in the effectiveness of the method. In addition, I noticed that the range of the spectral norm values that you used in your submission was very narrow (e.g., 5.0 - 6.0 in Table 13). Usually it is a good idea to sweep over a range of values with different order of magnitudes to get a better sense of how well the method does (e.g., 1.0, 10.0, 100.0). Having hyperparameter values too close to each other could increase the risk of overlooking good hyperparameters.
> >
> > >  Added experiments with additional baselines on DMControl environments
> >
> > In Table R1, it seems that LipsNet++ shows very little improvement over LipsNet-G and LipsNet-L (especially considering the confidence intervals). It is also very close to the MLP-SN baselines. Unfortunately, the experiments are not sufficiently strong to convince me that the approach is more effective compared to baselines.

---

> > > ### Author Response · Authors · 2025-04-06
> > >
> > > Thank you for your response and the new concerns.
> > >
> > > ---
> > > > What does $o(\delta)$ mean here?
> > >
> > > The term $o(\delta)$ refers to a **higher-order infinitesimal term** with respect to $\delta$ in the **Taylor expansion** (see Line 631 for the Taylor expansion process). This is a standard notation used in Taylor series. To improve clarity, we have updated Theorem 3.2 by explicitly writing *“where $o(\delta)$ is a higher-order infinitesimal term with respect to $\delta$.”*
> > >
> > > We appreciate your helpful suggestion, which indeed improves the readability of the theorem.
> > >
> > > > Tune the MLP-SN baselines with a similar budget as your method
> > >
> > > Please refer to Appendix F, where we have already provided a **thorough sensitivity analysis** for LipsNet++. LipsNet++ is **insensitive** to hyperparameters — it **only requires** selecting the **correct order** of magnitude without fine-tuning, and it **has only** low-sensitivity hyperparameters. In contrast, tuning MLP-SN requires carefully **adjusting the spectral norm** for **each layer** within a **narrow range**, which involves **highly sensitive** and **numerous hyperparameters combination**. Therefore, the tuning complexity of MLP-SN is not comparable to that of LipsNet++.
> > >
> > > We have added a discussion on the tuning complexity of MLP-SN in the revised paper — thank you for your suggestion, which helped improve our work.
> > >
> > > > Range of the spectral norm values was very narrow (e.g., 5.0 - 6.0). It is a good idea to sweep over different order of magnitudes (e.g., 1.0, 10.0, 100.0).
> > >
> > > Thank you for your careful reading and observation. We agree with your suggestion that sweeping over different orders of magnitude (e.g., 1.0, 10.0, 100.0) is a principled way to search for hyperparameters — in fact, we **adopted this approach where appropriate**, as shown in the hyperparameter tuning in Appendix F. However, we would like to kindly invite you to take a closer look at the results within the 5.0–6.0 range: from MLP-SN's **trends of TAR and AFR**, it is **already evident** that MLP-SN cannot outperform LipsNet++, as shown in the following Figure R3.
> > >
> > > Figure R3: [[Link - Figure R3]](https://github.com/ICML-anonymous-2025/LipsNet_v2/blob/main/rebuttal/Figure_R3.png)
> > >
> > > Following your suggestion, we also trained MLP-SN with spectral norm values in different orders of magnitude (e.g., 1.0, 10.0, 100.0), and the results are listed in the following Table R6.
> > >
> > > Table R6: [[Link - Table R6]](https://github.com/ICML-anonymous-2025/LipsNet_v2/blob/main/rebuttal/Table_R6.png)
> > >
> > > All the above results indicate that **no matter how** the hyperparameters of MLP-SN are **adjusted**—whether through fine-tuning within the range 5.0-6.0 or by coarser adjustments at the scale of 1, 10, or 100—its performance remains **significantly inferior to that of LipsNet++**.
> > >
> > > We believe this addition greatly helps readers better understand the comparison, and we sincerely appreciate your detailed and constructive feedback.
> > >
> > > > Shows very little improvement
> > >
> > > Please kindly note that the main goal of this paper is to **reduce the Action Fluctuation Rate (AFR)**. As shown in [[Link - Table R2]](https://github.com/ICML-anonymous-2025/LipsNet_v2/blob/main/rebuttal/Table_R2.png), LipsNet++ **achieves significant improvements** over the previous SOTA (LipsNet-L), **reducing AFR by 23.5%, 75.0%, 13.0%, and 35.5%** on Cartpole, Reacher, Cheetah, and Walker respectively, with an **average reduction of 36.74%**. These are **substantial improvements**. Moreover, the AFR **variances** of both LipsNet++ and LipsNet-L remain **below 0.03**, which is **much smaller** than the mean values, indicating that the **results are statistically reliable**. Compared to MLP-SN, the improvements are **even more pronounced**. These results **clearly demonstrate the SOTA performance of LipsNet++**.
> > >
> > > Thank you for raising this point — we have now included the AFR reduction percentages directly in the revised paper to more explicitly highlight the performance advantage of LipsNet++, which will help readers better appreciate the significance of results.
> > >
> > > ---
> > >
> > > We hope our reply adequately address your concern.
> > >
> > > All the above results have been incorporated into the revised paper. We must emphasize that your valuable feedback has significantly improved the paper's quality.
> > >
> > > Thank you sincerely for your time and valuable feedback!

---

### Official Review · Reviewer_QXnu · 2025-03-17

**Overall Recommendation:** 4

**Summary:**

Action fluctuation is a major issue in reinforcement learning. The fluctuation in action can be caused due to measurement noise or steep changes in the policy due to large Jacobians. This paper addresses the measurement noise issue using Fourier filter and steep Jacobian issue using Jacobian regularization. The approach is well-motivated and I believe the paper makes a strong contribution.

**Claims And Evidence:**

Yes, the exposition in the paper supports the claim in the introduction.

**Essential References Not Discussed:**

None that I know of.

**Experimental Designs Or Analyses:**

The experimental design appears valid to me, but I am more of a theoretically inclined researcher, I'll leave the critique on experiments to more experimentally oriented researchers.

**Methods And Evaluation Criteria:**

Yes from my perspective.

**Other Comments Or Suggestions:**

None.

**Other Strengths And Weaknesses:**

This paper is an excellent specimen of leveraging theoretical insights, especially using Lipschitz constant estimates to solve a crucial performance issue in RL.

**Questions For Authors:**

Isnt Lemma A.1 just the mean value theorem for vector valued functions? Any standard math textbook reference would suffice, e.g., see Principles of Mathematical Analysis by Rudin, Theorem 5.19.

**Relation To Broader Scientific Literature:**

The literature survey is sufficient to put the contribution in context.

**Theoretical Claims:**

Yes, the mathematical development is correct, but it can be made more rigorous. In the statement of Theorem 3.2, the authors should also include the approximation error for the Lipschitz constant instead of just using $\approx$ notation, to make the theorem more precise. The approximation error appears well articulated in the proof in the appendix, so suggest just using that formulation.

---

> ### Author Rebuttal · Authors · 2025-03-31
>
> Thank you for your insightful feedback!
> We appreciate your recognition of our work as a ***well-motivated*** approach that makes a ***strong contribution*** and as an ***excellent specimen of leveraging theoretical insights to solve a crucial issue in RL***. Your positive assessment greatly encourages us.
>
> We now address your comments in detail:
>
> ---
> #### **> Theoretical Improvement**
> We appreciate your valuable suggestion! We **have refined** Theorem 3.2 by **explicitly incorporating the approximation error and its asymptotic behavior**. The revised theorem is as follows:
>
> **Theorem 3.2** (Lipschitz’s Jacobian Approximation): *Let $f:\mathbb{R}^n \to \mathbb{R}^m$ be a continuously differentiable neural network. The Jacobian norm $\left\Vert\nabla_xf\right\Vert$ serves as an approximation of the local Lipschitz constant of $f$ within the neighborhood $\mathcal{B}(x, \rho)$, centered at $x$ with radius $\rho$. The approximation error is given by $\mathop{\max}_{\ \delta\in \mathcal{B}(0,\rho)} \left[ \left(\nabla_x\left\Vert \nabla_x f(x) \right\Vert\right)^\top\delta + o(\delta)\right].$ Moreover, as $\rho \to 0$, the Jacobian norm converges to the exact local Lipschitz constant, i.e. $\left\Vert\nabla_xf\right\Vert \to K(x)$.*
>
> This refinement provides a more precise characterization of the approximation error, as you suggested.
>
> #### **> Question on Lemma A.1**
> We appreciate your question regarding Lemma A.1. Theorem 5.19 in Rudin's book is an **existential result** that relates the function’s increment to its derivative at some point, **without involving** the Lipschitz constant or providing an explicit expression for Lipschitz constant; whereas Lemma A.1 **explicitly gives an equivalent expression** for the Lipschitz constant. In summary, Theorem 5.19 primarily focuses on the bounding variation of a function, while Lemma A.1 focuses on the local Lipschitz property and provides an approximation method for calculating the Lipschitz constant.
>
> ---
> #### **Summary**
> We hope our revisions adequately address your concerns. Please let us know if you have any further questions.
>
> Thank you again for your time and valuable feedback!

---

### Official Review · Reviewer_Exvu · 2025-03-17

**Overall Recommendation:** 3

**Summary:**

The paper introduces LipsNet++, a novel policy network designed to mitigate action fluctuation in reinforcement learning (RL). The authors identify two primary causes of action fluctuation: observation noise and policy non-smoothness. To address these, LipsNet++ integrates: A Fourier filter layer, which processes historical observations in the frequency domain and applies a trainable filter matrix to suppress noise. And, a Lipschitz controller layer, which applies Jacobian regularization to constrain the Lipschitz constant, ensuring smooth control outputs.


## update after rebuttal
The authors did address most of my concerns, which is why I increased my score.

**Claims And Evidence:**

The paper presents several claims, the majority of which are well-supported by experimental results:

- The Fourier filter layer effectively suppresses observation noise.
- The Lipschitz controller layer enhances policy smoothness, supported by theorem 3.2.
- The approach is applicable across various continuous control tasks.
- LipsNet++ is easily integrable into different network architectures and RL frameworks.

**Essential References Not Discussed:**

The related work section appears comprehensive, and I did not identify any missing references.

**Experimental Designs Or Analyses:**

The experiments are well-structured, incorporating multiple environments and ablations:
- Double Integrator: Assesses basic control performance and robustness to noise.
- DMControl Suite: Evaluates performance on complex continuous control tasks.
- Mini-Vehicle Driving: Demonstrates real-world applicability in a robotics setting.

It would have been interesting to examine the impact of control type, as position control inherently applies some degree of smoothing compared to torque control. A useful addition would be to construct two alternative control interfaces (for position and velocity; DMC already uses torque) within DMC and perform an ablation study over them. Given the exploratory nature of this analysis, it may not be necessary to compare all baselines for this ablation.

**Methods And Evaluation Criteria:**

Yes, the proposed methods and evaluation criteria make sense. However, one aspect that is missing is the influence of control type. Depending on the environment, the control inputs could be torque, velocity, or position-based. It would be particularly interesting to investigate how LipsNet++ performs across these different control modalities and whether its effectiveness varies between them.

**Other Comments Or Suggestions:**

- second formula: squared term needs to be inside the expectation
- “3.1. Reasons Identification of Action Fluctuation” not a good section name. I also found that section not really novel and satisfying. Calling it one of the four core contributions of the paper looks like an overstatement to me

**Other Strengths And Weaknesses:**

Strengths:
- Novel integration of filtering and control into a policy network.
- Clear empirical improvements in smoothness and robustness.
- Good ablation studies.
- Public PyTorch implementation for reproducibility.

Weaknesses:
- Limited discussion on scalability to image-based RL.
- Some interesting ablations are missing (see above on control type influence)
- Certain claimed contributions seem overstated relative to their novelty.

**Questions For Authors:**

- how well does this scale to image observations?
- does the trainable H matrix receive gradients from the critic? Or did you stop these gradients?

**Relation To Broader Scientific Literature:**

The paper builds on prior work in:
- Lipschitz-constrained policy learning (Takase et al., 2020; Song et al., 2023).
- Fourier-based filtering in neural networks (Lee-Thorp et al., 2022; Tan et al., 2024).
- Smooth control in RL (Mysore et al., 2021; Yu et al., 2021).

The connection to classical control theory is a valuable insight, suggesting that RL policy networks can explicitly incorporate filtering and control mechanisms.

**Theoretical Claims:**

I did take a look at theorem 3.2. The derivations are mathematically sound and align with prior work on smooth policy learning.

---

> ### Author Rebuttal · Authors · 2025-04-01
>
> Thank you for your insightful feedback!
> We are encouraged by the positive aspects of your assessment, i.e., ***well-supported*** claims, ***mathematically sound*** derivations, ***valuable*** insight, ***comprehensive*** related works, ***novel*** integration of filtering and control, ***clear*** empirical improvements, and ***good*** ablation studies.
>
> We now address your comments in detail:
>
> ---
> #### **> Examine Impact of Control Types**
> All DMControl environments in our paper (Cartpole, Reacher, Cheetah, Walker) are torque control. To examine the impact of control types as you suggested, we additionally train on **position control (DMControl Fish)** and **velocity control (Gymnasium Bipedal Walker)** environments. The Fish task needs to control 5 actions (tail, tail_twist, fins_flap, finleft_pitch, finright_pitch) under position control type, and the Bipedal Walker needs to control 4 motor speed values under velocity control type. Their TAR and ARF are summarized as follows:
>
> TAR comparison: [[Link - Table R3]](https://github.com/ICML-anonymous-2025/LipsNet_v2/blob/main/rebuttal/Table_R3.png), AFR comparison: [[Link - Table R4]](https://github.com/ICML-anonymous-2025/LipsNet_v2/blob/main/rebuttal/Table_R4.png)
>
> These results show that LipsNet++ achieves good action smoothness across all three control types. Additionally, some studies on robotics control also support the effectiveness of LipsNet-related structures in torque control [1] and position control [2].
>
> In fact, we believe LipsNet++ does not assume the control type to be torque, position, or velocity; it ensures action smoothness in all cases and serves as a general solution. The choice of control type should depend on the task requirements.
>
> #### **> Other Comments Or Suggestions**
> - Formula in Line 98: Typo fixed.
> - Thank you for your suggestion. We **have renamed** Section 3.2 to “Understanding Key Factors Behind Action Fluctuation” and **refined** the contribution description accordingly. Additionally, we believe analyzing the root causes of action fluctuation is essential. Equation (1) mathematically identifies two fundamental causes, which play a crucial role in guiding the design of the subsequent network layers (Fourier Filter Layer & Lipschitz Controller Layer). Previous works did not distinguish the two causes and treated them as one; therefore, our analysis actually contributes to the broader community.
>
> #### **> Q1: Scalability to Image-based RL**
> **Yes**, LipsNet++ can scale to image observations. The **core question** here is whether LipsNet++ can handle high-dimensional observation tasks. To demonstrate this, we added two experiments: DMControl Cartpole (pixel observation) and Gymnasium Humanoid (348 observations), with the former has image-based observations and the latter has high-dimensional observations.
>
> Given the high-dimensional input, we adopt the following LipsNet++ architecture:
> - DMControl Cartpol: *[Convolutional Layer -> Fourier Filter Layer -> Lipschitz Controller Layer]*
> - Gymnasium Humanoid: *[Linear Layer -> Fourier Filter Layer -> Lipschitz Controller Layer]*
>
> In the Convolutional Layer and Linear Layer, they perform the feature extraction and dimension reduction.
>
> The TAR and AFR are summarized in [[Link - Table R5]](https://github.com/ICML-anonymous-2025/LipsNet_v2/blob/main/rebuttal/Table_R5.png). These results demonstrate that **LipsNet++ can handle high-dimensional observation tasks and can scale to image-based tasks**.
>
> #### **> Q2: Gradient Flow for Matrix $H$**
> $H$ receives gradients from the policy improvement loss $\mathcal{L}'$, which includes the critic value as described in Equation (4) and the last equation in Section 2.1; but $H$ does not directly receive gradients from the critic itself, as the critic does not have a Fourier Filter Layer.
>
> Integrating a Fourier Filter Layer into the critic is an interesting direction. In this case, the actor and critic could potentially share one $H$ matrix, raising an interesting question of whether $H$ should receive gradients only from the actor or from both. Thank you for your insightful perspective. We will explore this in future work.
>
> ---
> #### **Summary**
> We hope our revisions adequately address your concerns. Please let us know if you have any further questions.
>
> Thank you again for your time and valuable feedback!
> #### ***References***
> *[1] Y. Zhang, et al. Robust Locomotion Policy with Adaptive Lipschitz Constraint for Legged Robots. IEEE RAL, 2024.*
>
> *[2] G. Christmann, et al. Benchmarking Smoothness and Reducing High-Frequency Oscillations in Continuous Control Policies. IEEE IROS, 2024.*

---

> > ### Comment · Reviewer_Exvu · 2025-04-05
> >
> > I sincerely thank the authors for their detailed response and for taking the time to run the experiment I requested. I truly appreciate their effort. Most of my concerns have been addressed, and as a result, I will be raising my score.

---

### Decision · Program_Chairs · 2025-05-01

**Decision:**

Accept (spotlight poster)

**Comment:**

This paper proposes a novel policy network, LipsNet++, that aims to address the action fluctuation problem in RL by integrating a Fourier filter layer (to address observation noise) and a Lipschitz controller layer (to ensure policy smoothness). In experiments, the paper demonstrates convincing improvements and/or on-par results for total average return, while yielding significant improvements in action fluctuation ratio, on DMControl benchmark tasks compared to the previous SOTA. Physical experiments on mini-vehicles are also implemented.

While several reviewers initially raised concerns about the rigor of the theoretical analysis, the generalizability of experimental settings, and the number of baselines compared against, and the potential effect of noise, the authors were largely able to address these concerns via updated analysis and additional experiments during the rebuttal period. Overall, the reviewers largely agreed that this is a valuable contribution.